# BEYOND SEQUENCE: IMPACT OF GEOMETRIC CONTEXT FOR RNA PROPERTY PREDICTION

**Junjie Xu**[1,2] [*][†]**, Artem Moskalev**[1] [†]**, Tommaso Mansi**[1]**, Mangal Prakash**[1] [‡]**, Rui Liao**[1] [‡]

[1]Johnson & Johnson Innovative Medicine, [2]The Pennsylvania State University
`junjiexu@psu.edu`
`{amoskal2, tmansi, mpraka12, rliao2}@its.jnj.com`

## ABSTRACT

Accurate prediction of RNA properties, such as stability and interactions, is crucial for advancing our understanding of biological processes and developing RNA-based therapeutics. RNA structures can be represented as 1D sequences, 2D topological graphs, or 3D all-atom models, each offering different insights into its function. Existing works predominantly focus on 1D sequence-based models, which overlook the geometric context provided by 2D and 3D geometries. This study presents the first systematic evaluation of incorporating explicit 2D and 3D geometric information into RNA property prediction, considering not only performance but also real-world challenges such as limited data availability, partial labeling, sequencing noise, and computational efficiency. To this end, we introduce a newly curated set of RNA datasets with enhanced 2D and 3D structural annotations, providing a resource for model evaluation on RNA data. Our findings reveal that models with explicit geometry encoding generally outperform sequence-based models, with an average prediction RMSE reduction of around 12% across all various RNA tasks and excelling in low-data and partial labeling regimes, underscoring the value of explicitly incorporating geometric context. On the other hand, geometry-unaware sequence-based models are more robust under sequencing noise but often require around $2 - 5\times$ training data to match the performance of geometry-aware models. Our study offers further insights into the trade-offs between different RNA representations in practical applications and addresses a significant gap in evaluating deep learning models for RNA tasks.
All datasets and code will be released upon acceptance.

## 1 INTRODUCTION

RNA plays a central role in the machinery of life, serving as a crucial intermediary between nucleotide and amino acid worlds (Sahin et al., 2014). Beyond its messenger role, RNA is involved in diverse biological processes, including gene regulation, catalytic activity, and structural support within ribosomes (Sharp, 2009; Strobel et al., 2016). This versatility makes RNA a key target for fundamental biological research and therapeutic interventions. As our understanding of RNA complexity grows, so does the need for advanced computational tools for its analysis.

Modeling RNA is challenging due to its intricate secondary and tertiary structures, dynamic conformational changes, and interactions with cellular components (Han et al., 2015). Furthermore, RNA analysis is hindered by the practical challenges of RNA data acquisition which include sequencing errors (Ozsolak & Milos, 2011), batch effects (Tran et al., 2020), incomplete sequencing (Alfonzo et al., 2021), partial labeling (Wayment-Steele et al., 2022b), and high costs of obtaining large labeled datasets (Byron et al., 2016). Moreover, RNA molecule can be represented in different ways: as a 1D nucleotide sequence, a 2D graph of base pairings, or a 3D atomic structure. Each representation highlights different aspects of RNA, presenting both opportunities and challenges for model design and selection (Fig. 1).

---

[*]This work was done while the author was an intern at Johnson & Johnson.
[†]Equal contribution as first authors
[‡]Equal contribution as last authors

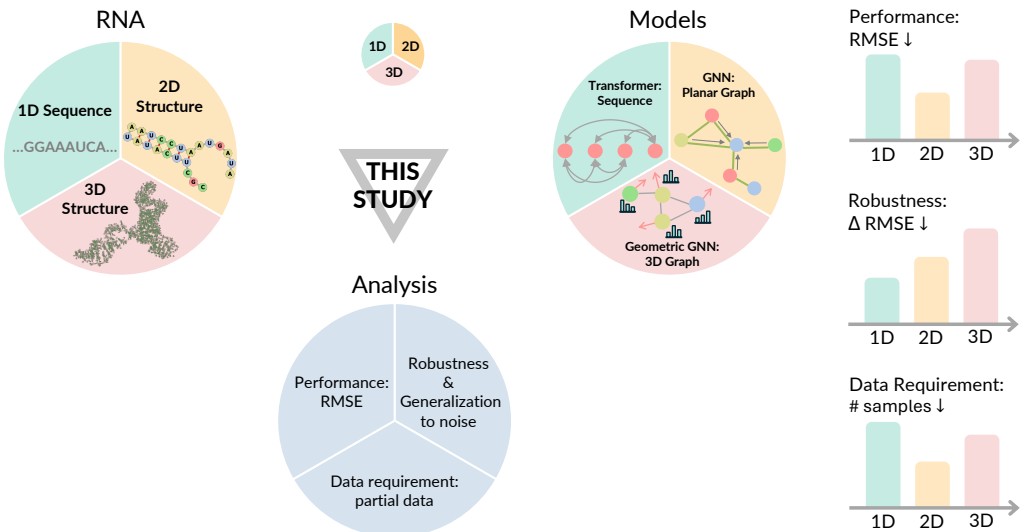

Figure 1: **Overview of the study.** (a) Left panel: RNA sequences represented in 1D, 2D, and 3D structures, processed by 1D sequence, 2D GNN, and 3D GNN models. Our analysis includes prediction error, robustness and generalization to sequencing noise, and performance under limited training data and partial labelings. (b) Right panel: Comparative performance of 1D, 2D, and 3D methods across experimental conditions. Histograms show RMSE performance, relative RMSE changes with increasing noise, and data requirements for optimal performance. Lower values indicate better performance in all metrics.

In this work, we systematically study the performance of various machine learning models for RNA property prediction, extending beyond traditional sequence-based approaches (He et al., 2021; Soylu & Sefer, 2023; Prakash et al., 2024; Yazdani-Jahromi et al., 2025) to include methods that process RNA with its 2D or 3D geometry. While 2D and 3D RNA representations offer potentially richer information, they also present unique challenges. In the absence of high-quality experimental data, accessing 2D or 3D RNA structures requires running structure prediction algorithms prone to noise and mistakes, especially in the presence of sequencing errors (Schneider et al., 2023; Wang et al., 2023). A few mutations in the nucleotide sequence owing to sequencing mistakes can hugely alter the 2D and 3D structure (Fig. 4), potentially undermining the benefit of using additional geometric context. Furthermore, real-world RNA datasets often suffer from partial labeling (Wayment-Steele et al., 2022b) and scarcity of training samples (Wint et al., 2022), which may affect geometry-aware methods differently than sequence-based approaches. These challenges raise a fundamental question: to what extent does explicit 2D and 3D geometry contribute to RNA property prediction, and under what circumstances might it offer advantages over geometry-free sequence models?

In this study, we seek to answer this question by making the following contributions:

- We introduce a diverse collection of RNA datasets, including newly annotated 2D and 3D structures, covering various prediction tasks at nucleotide and sequence levels across multiple species.
- We provide a unified testing environment to evaluate different types of machine learning models for RNA property prediction, including sequence models for 1D, graph neural networks for 2D, and equivariant geometric networks for 3D RNA representations.
- We conduct a comprehensive analysis of how different models perform under various conditions, such as limited data and labels, different types of sequencing errors, and out-of-distribution scenarios. We highlight the trade-offs and contexts in which each modeling approach is most effective, guiding researchers in selecting suitable models for specific RNA analysis challenges.
- We also introduce novel modifications to existing 3D geometric models based on biological prior, specifically optimizing them for handling large-scale point cloud RNA data, thus improving the efficiency and performance of 3D models significantly.

Our study reveals several key insights: **(i)** 2D models generally outperform 1D models, with spectral GNNs reducing prediction error by about 12% on average across all datasets, highlighting the importance of explicitly considering RNA structural information; **(ii)** 3D equivariant GNNs outper-

form 1D and some 2D methods in noise-free scenario but are sensitive to noise, exhibiting up to a 56% decrease in prediction quality under high sequencing error rates; **(iii)** geometry-free sequence models remain the most robust to sequencing noise, showing only a 14-27% increase in prediction error compared to noise-free conditions, however they require around $2 - 5\times$ more training data to match the performance of geometry-aware models.

## 2 DATASETS AND MODELS

Here, we discuss datasets selected for our study with RNA-level prediction labels. These datasets are selected to vary from small to large-scale and to encompass both nucleotide-level tasks and sequence-level tasks. We perform an extensive evaluation across these datasets, leveraging three different model families (1D, 2D, 3D) spanning 9 representative models in total.

### 2.1 DATASETS

The datasets vary in size based on the number of sequences and sequence lengths: the small dataset Tc-Riboswitches (Groher et al., 2018), the medium datasets Open Vaccine COVID-19 (Wayment-Steele et al., 2022b) and Ribonanza-2k (He et al., 2024), and the large dataset Fungal (Wint et al., 2022). All datasets provide regression labels. Detailed statistics for these datasets are provided in Appendix F.1.

1. **Tc-Riboswitches**: 355 mRNA sequences (67-73 nucleotides) with sequence-level labels for tetracycline-dependent riboswitch switching behavior, important for optimizing gene regulation in synthetic biology and gene therapy.

2. **Open Vaccine COVID-19**: 4,082 RNA sequences (each of 107 nucleotides) with nucleotide-level degradation rate labels, crucial for predicting RNA stability in mRNA vaccine development.

3. **Ribonanza-2k**: 2,260 RNA sequences (each of 120 nucleotides) with nucleotide-level experimental reactivity labels, supporting RNA structure modeling and RNA-based drug design.

4. **Fungal**: 7,056 coding and tRNA sequences (150-3,000 nucleotides) from 450 fungal species, used for sequence-level protein expression prediction.

### 2.2 DATA PREPROCESSING AND CURATION

For the OpenVaccine COVID-19 dataset, we filter out sequences with a signal-to-noise ratio (SNR) below 1, as recommended by the dataset authors (Wayment-Steele et al., 2022b), to ensure that only sequences with a significant signal relative to background noise are included, thereby enhancing the reliability of modeling. For the other datasets, we use the original sequences since no SNR annotations are available.

Since all the RNA datasets come with sequences only, we employ EternaFold (Wayment-Steele et al., 2022a) and RhoFold (Shen et al., 2022) to infer 2D and 3D molecular structures respectively. We selected EternaFold and RhoFold due to their state-of-the-art performances acknowledged in recent works (Wayment-Steele et al., 2020; 2022b; He et al., 2024) Additionally, RhoFold typically runs in seconds to a minute per sequence, unlike other 3D structure prediction tools which usually take hours, and hence not suitable for large datasets.

For 1D modeling, we use the original RNA sequences without structural augmentation which equates to processing a plain string of nucleotides. The 2D datasets represent each RNA sequence as a graph with nodes for nucleotides and edges for bonds between nucleotides. The node features are six-dimensional, incorporating one-hot nucleotide identity ('A', 'C', 'G', 'U') alongside the sum and mean base-pairing probabilities (BPP), which are available from 2D structure prediction tools. In 3D, each RNA molecule is represented as a graph, with nodes corresponding to individual atoms. Node features represent one-hot atom identity.

### 2.3 MODELS

We select well-established model architectures recognized for their state-of-the-art performance for molecular property prediction tasks in various domains. **1D Model**: Transformer1D (Honda et al.,

2019; He et al., 2021); RNA-FM (Chen et al., 2022), SpliceBERT (Chen et al., 2023) **2D Models**: GCN (Kipf & Welling, 2017; Wieder et al., 2020), GAT (Veličković et al., 2018; Ye et al., 2022), ChebNet (Defferrard et al., 2016; Knyazev et al., 2018), Transformer1D2D (He et al., 2023), Graph Transformer (Shi et al., 2020; Li et al., 2022), and GraphGPS (Rampášek et al., 2022; Zhu et al., 2023); **3D Models**: SchNet (Schütt et al., 2017; Han et al., 2022), EGNN (Satorras et al., 2021), FAENet (Duval et al., 2023), DimeNet (Gasteiger et al., 2020), GVP (Jing et al., 2020), and FastEGNN (Zhang et al.). Detailed descriptions of all the models can be found in Appendix Sec. B.

**Training and evaluation**     All models were trained on a NVIDIA A100 GPU. To ensure hyperparameter parity for each baseline, hyperparameters were optimized using Optuna (Akiba et al., 2019), restricting the search to models with fewer than 10 million parameters that fit within the GPU memory constraint of 80GB. All model hyperparameters, training, and evaluation details are reported in Appendix H. We ran all models for 5 random data splits (train:val:test split of 70:15:15) and we report average performance with a standard deviation across splits. The mean column-wise root mean squared error (MCRMSE), introduced in Wayment-Steele et al. (2020), is used as the evaluation metric. It is defined as $\text{MCRMSE}(f, D) = \sqrt{\frac{1}{n} \sum_{i=1}^{n} (\hat{y}_i - y_i)^2}$, where $f$ represents a model, $D$ is the dataset, and $\hat{y}_i$ and $y_i$ are the predicted and true values for data point $i$.

## 3    Task Definitions

Here, we introduce the downstream tasks for evaluating models for RNA property prediction. Each task is designed to quantify specific behaviors under various real-world experimental conditions.

**Task 1: Impact of structural information on prediction performance**     This task aims to evaluate how incorporating RNA structural information affects prediction quality. We compare the performance of models using 1D (sequence-only), 2D, and 3D RNA representations to determine if and to what extent geometric data improves property prediction.

**Task 2: Model efficiency in limited training data settings**     Acquiring high-quality comprehensive RNA datasets is often challenging and resource-intensive thus limiting the amount of labeled data for training (Teufel & Sobetzko, 2022; Byron et al., 2016). This task aims to investigate how model performance depends on the amount of training data used, evaluating the sample efficiency of each family of models. In other words, given a dataset $D = \{X, Y\}$, let $D_\alpha = \{X_\alpha, Y_\alpha\}$ be a subset where the training set is reduced to a fraction $\alpha$. We train models on different sets of $D_\alpha$ datasets with decreasing $\alpha$.

**Task 3: Performance with partial sequence labeling**     Due to the high cost of measuring properties for every nucleotide in RNA sequence, real-world datasets often contain partial annotations (Wayment-Steele et al., 2022b) where labels are only available for the first small part of the sequence. This task is relevant for nucleotide-level datasets and it aims to investigate how well a model can generalize to a whole RNA sequence when labels are only available for a portion of it.

**Task 4: Robustness to sequencing noise**     Acquiring RNA data requires sequencing. In practice sequencing procedure may introduce sequencing errors (random mutations of nucleotides) that vary depending on the sequencing technology and platform (Ozsolak & Milos, 2011; Fox et al., 2014). These errors affect the raw sequence data, and propagate to structural noise in 2D and 3D. The goal of this task is to assess how well models can maintain reliable performance when trained and tested under the same distribution of realistic levels of sequencing noise observed in practice, ensuring robustness across a consistent noise environment. This reflects real-world cases where a specific sequencing method produces noisy data, but the noise characteristics are stable across training and deployment.

**Task 5: Generalization to Out-of-Distribution (OOD) data**     This task focuses on a different practical challenge: models trained on high-quality RNA sequences are often deployed in conditions where the data exhibits different noise characteristics due to batch effect (Tran et al., 2020) or the use of different sequencing platforms (Tom et al., 2017). Here, the objective is to evaluate how well models generalize to OOD datasets with different levels of sequencing noise, assessing the

Table 1: **Comparison of 1D, 2D, and 3D models across datasets. Bold** indicates the best, underline the second-best. 'OOM' means out-of-memory. ChebNet excels by capturing global graph information. Overall, 2D models outperform 1D models, highlighting the value of structural information. Although 3D models face challenges with scalability and noisy predictions, our nucleotide pooling strategy, based on biological prior, enhances their performance on shorter sequences, allowing 3D encoding to occasionally surpass 1D models. See Sec. 4.1 for details on nucleotide pooling strategy.

| Model | COVID | Ribonanza | Tc-Ribo | Fungal |
|---|---|---|---|---|
| *1D model* | | | | |
| Transformer1D | 0.361±0.017 | 0.705±0.015 | 0.705±0.079 | 1.417±0.005 |
| RNA-FM | 0.591±0.081 | 0.990±0.144 | 0.693±0.001 | 1.420±0.028 |
| SpliceBERT | 0.588±0.077 | 1.022±0.144 | 0.708±0.003 | 1.435±0.059 |
| *2D model* | | | | |
| Transformer1D2D | 0.305±0.012 | 0.514±0.004 | 0.633±0.001 | OOM |
| GCN | 0.359±0.009 | 0.595±0.006 | 0.701±0.004 | 1.192±0.077 |
| GAT | 0.315±0.006 | 0.534±0.006 | 0.685±0.024 | 1.112±0.035 |
| ChebNet | **0.279±0.007** | **0.468±0.002** | **0.621±0.022** | **0.973±0.003** |
| Graph Transformer | 0.318±0.008 | 0.515±0.001 | 0.710±0.041 | 1.317±0.002 |
| GraphGPS | 0.332±0.013 | 0.523±0.003 | 0.715±0.012 | 1.025±0.081 |
| *3D model (without pooling)* | | | | |
| EGNN | 0.480±0.025 | 0.808±0.023 | 0.725±0.002 | OOM |
| SchNet | 0.499±0.003 | 0.843±0.004 | 0.696±0.008 | OOM |
| FAENet | 0.486±0.010 | 0.834±0.003 | 0.703±0.011 | OOM |
| DimeNet | 0.497±0.012 | 0.855±0.006 | 0.712±0.004 | OOM |
| GVP | 0.467±0.010 | 0.797±0.012 | 0.744±0.004 | OOM |
| FastEGNN | 0.477±0.005 | 0.816±0.014 | 0.753±0.001 | OOM |
| *3D model (with nucleotide pooling)* | | | | |
| EGNN (pooling) | 0.364±0.003 | 0.619±0.007 | 0.663±0.010 | OOM |
| SchNet (pooling) | 0.390±0.006 | 0.685±0.006 | 0.655±0.038 | OOM |
| FastEGNN (pooling) | 0.444±0.003 | 0.753±0.015 | 0.710±0.011 | OOM |

extent of performance degradation as noise levels increase. This task simulates the scenario where a model encounters noisier data than it was trained on, highlighting its ability to adapt to unexpected experimental conditions.

For detailed descriptions and motivations of the five task settings, please refer to Appendix D.

## 4 EXPERIMENTS AND RESULTS

### 4.1 IMPACT OF EXPLICIT GEOMETRY LEARNING ON MODEL PERFORMANCE

We begin by addressing Task 1, where we compare the performance of model families when trained and evaluated on the downstream RNA datasets. Additionally, we provide runtime and memory comparison in Appendix C.

**2D models consistently outperform 1D model** Results in Table 1 reveal that 2D methods consistently outperform the 1D sequence model across all datasets. Notably, the Transformer1D2D model, which simply augments the attention matrix with adjacency features alongside, achieves around 10% lower prediction MCRMSE on average across datasets than its geometry-free counterpart. This suggests that explicitly incorporating structural information is crucial, as learning from sequence data alone proves to be insufficient. Further experiments, detailed in the Appendix F.4, investigate the learned attention maps of both the Transformer1D2D and the Transformer1D model and their correlation with structural information and reveal that Transformer1D2D attention maps are much more closely aligned with the topological structure of nucleotide graph, reinforcing the conclusion that explicit encoding of structural information is essential for improved performance.

For the foundation models, RNA-FM and SpliceBERT, we observe that for 2 out of 4 datasets (Covid and Ribonanza-2k), RNA foundation models perform worse than the simple supervised transformer baseline whereas for the other two datasets (Tc-Riboswitches and Fungal), transformer and RNA foundation models achieve similar performance. This is consistent with recent works in multiple biology-related domains demonstrating specialized foundation models are yet to surpass simple supervised learning baselines (Xu et al., 2024b; Kedzierska et al., 2023; Yang et al., 2024). Hence, we choose Transformer1D as the model of choice for subsequent experiments.

**Spectral GNN outperforms spatial GNNs in 2D** ChebNet, a spectral method, outperforms spatial methods such as GCN, GAT, Graph Transformer, and GraphGPS, achieving a prediction MCRMSE 2.5% lower than the next best 2D model across datasets. Spatial GNNs aggregate node features layer by layer, emphasizing local information within a fixed distance. While computationally efficient, these methods are limited by the 1-Weisfeiler-Lehman (WL) test, which constrains the expressive power of node-based updates (Xu et al., 2018). In addition, spatial GNNs may suffer from a limited receptive field while spectral methods approximate global graph features, enabling a global receptive field since the first layer Wang & Zhang (2022); Bo et al. (2023); Xu et al. (2024a). This allows ChebNet to effectively process global information which is important for RNA data due to potential long-range interaction between nucleotides.

**Challenges of modeling geometric context in all-atom resolution** Contrary to our expectations, 3D models at all-atom resolution (EGNN w/o pooling, SchNet w/o pooling DimeNet (w/o pooling), FAENet (w/o pooling), GVP (w/o pooling) and FastEGNN (w/o pooling) in Table 1) show relatively high prediction MCRMSE across datasets, underperforming compared to 1D and 2D methods. We hypothesize this is due to two factors.

First, all-atom 3D models rely on a limited local neighborhood of adjacent atoms, limiting their receptive fields and preventing them from capturing long-range dependencies (see Appendix Table F.1), which can be crucial for determining RNA properties (Shetty et al., 2013; Alshareedah et al., 2019). Expanding the local neighborhood for these methods becomes challenging due to the overwhelming scale of large molecular systems in all-atom resolution. Second, the performance of 3D models is often limited by the inherent inaccuracies in 3D structure prediction tools, which are generally less reliable compared to 2D structure prediction methods (Ponce-Salvatierra et al., 2019). While FastEGNN is designed for larger molecules, its reliance on the center of mass virtual node initialization may not align well with tasks like RNA property prediction, which inherently have a sequence prior. This may explain the limited performance of FastEGNN in this context and RNA-specific virtual node initialization and carefully designed pooling mechanisms may be needed to further adopt this architecture for large-molecules with a sequence prior.

To address the receptive field limitation of all-atom methods, we employ a biological prior by pooling atomic features into nucleotide-level representations after a few layers of all-atom operations. This strategy is aligned with RNA's natural secondary structure, where atoms group into nucleotides, and nucleotides form the complete RNA molecule (Deng et al., 2023). This novel strategy allows us to maintain all-atom resolution in the initial layers while increasing the receptive field via pooling. By balancing the number of all-atom and nucleotide layers as hyperparameters, we can balance fine- and coarse-grained all-atom and nucleotide resolution. Compared to other 3D models, EGNN, FastEGNN, and SchNet perform slightly better, and hence we introduce the nucleotide pooled variants for these models. We call 3D models with pooling EGNN (nuc. pooling), FastEGNN (nuc. pooling), and SchNet (nuc. pooling). This strategy significantly enhances 3D model performance, reducing prediction MCRMSE by ∼10% compared to the original EGNN, SchNet, and FastEGNN and outperforming 1D models on the Ribonanza-2k and Tc-Riboswitches datasets by 5% on average. On the COVID dataset, EGNN (nuc. pooling) matches the Transformer1D model but still trails behind 2D models. All subsequent experiments report results using nucleotide-pooled versions of the 3D models.

Next, we investigated the second hypothesis regarding higher noise in 3D structures by quantifying variability in predicted structures across different 3D prediction tools in Appendix G. We observe substantial variability (between 11-45Å RMSD across structures given by different 3D structure prediction tools), suggesting considerable noise in 3D predictions, which likely contributes to the poorer performance of 3D models.

## 4.2 MODEL EFFICIENCY UNDER LIMITED DATA AND PARTIAL SEQUENCE LABELING

In this section, we combine the analysis of Tasks 2 and 3, assessing model performance in scenarios with limited training data or partial labels.

To analyze how the amount of training data influences model performance, we run experiments with varying portions of the full datasets (25%, 50%, 75%, and 100%) on the medium- and large-scale datasets: COVID, Ribonanza-2k, and Fungal (Appendix Fig. 8(a) for illustration). The small size of the Tc-Riboswitches dataset is excluded from the analysis, as training with lower ratios would have resulted in inadequate sample sizes for meaningful evaluation. Additionally, GPU memory constraints prevent the application of Transformer1D2D and 3D models on the Fungal dataset due to its large sequence length.

We also evaluate the impact of partial property labels for nucleotide-level tasks, a common occurrence owing to costly experimental measurements (Wayment-Steele et al., 2020) to identify which models are best suited to handle the challenges of incomplete labels in RNA property prediction. For this, we use the COVID and Ribonanza datasets as these datasets contain nucleotide-level labels. We train the models using all training data but with varying proportions of labeled nucleotides (20%, 40%, 60%, 80%, and 100%) per sequence, thus simulating incomplete or sparse labeling, while testing on fully labeled test sets (Appendix Fig. 8(b) for illustration).

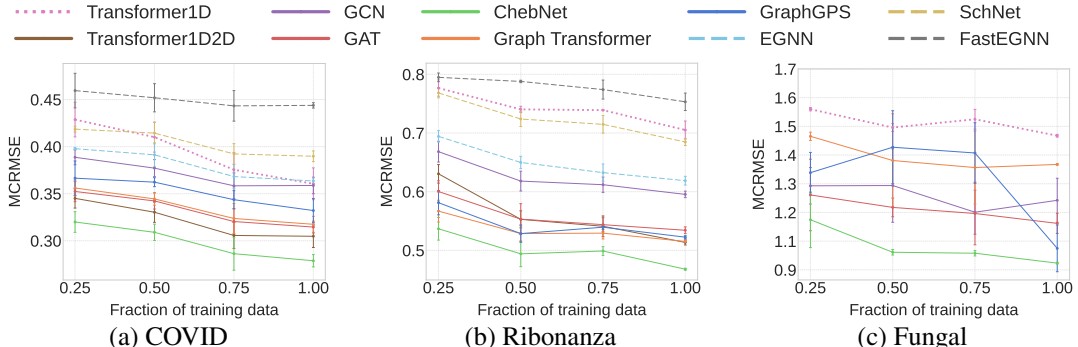

Figure 2: **Performance vs. fraction of training data across various datasets.** Model performance improves with increasing data, with lower MCRMSE across all models. 2D models consistently outperform 1D models, particularly in low-data regimes, underscoring the value of structural information for generalization. Dotted, solid, and dashed lines denote 1D, 2D, and 3D methods, respectively, which applies consistently throughout all figures in this paper.

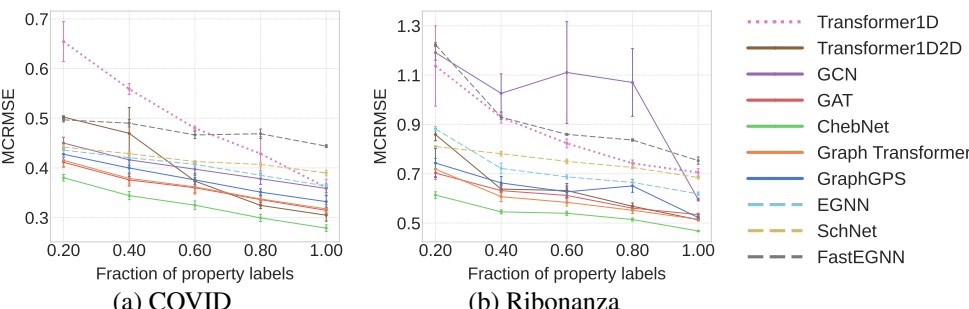

Figure 3: **Performance vs. partial property labels on COVID and Ribonanza datasets.** 2D models consistently outperform 1D models with sparse labeling, while Transformer1D and Transformer1D2D improve rapidly with denser supervision, emphasizing the need for more labels in transformer-based models.

**More training data improves performance** Unsurprisingly, across all models and datasets, a clear trend emerges: increasing the amount of available data, whether through higher number of training data points or greater proportion of available labels leads to improved performance (Fig. 2, Fig. 3, Appendix Tables in Sec. I.1, I.2). However, the degree of performance improvement varies significantly between model types, as analyzed next.

**2D models excel in low data and partial label regimes** Evaluating model performance at different training and label ratios reveals a notable trend: 2D models consistently outperform 1D and 3D models under low data and incomplete labeling regimes. Between 20-50% training data and label levels, 2D models such as ChebNet, Transformer1D2D, GraphGPS, and GAT significantly outperform Transformer1D, highlighting the role of additional structural information for model sample efficiency. Interestingly, Transformer1D and Transformer1D2D exhibit a faster rate of improvement when more labels are available (Fig. 3), suggesting that transformer-based architectures benefit from denser supervision. Notably, Transformer1D requires $2 - 5\times$ more training data/labels to match the performance of the least effective 2D models, which achieve comparable results using only 20% to 50% of the training data needed by Transformer1D when trained on the full dataset.

**3D models outperform 1D model in limited data regime despite structural noise** For the medium-scale datasets (COVID and Ribonanza), where 3D models can be evaluated, we observe that the 3D models generally outperform or are on par with the Transformer1D, even for lower data and labeling regimes. This suggests that despite the noise introduced by inaccuracies in 3D structure predictions, the explicit geometric encoding in 3D models still provides an advantage over 1D models. EGNN, in particular, is consistently better than or on par with Transformer1D across all training and label ratios. This further emphasizes that models incorporating explicit geometric encoding (whether 2D or 3D) are more data-efficient than those relying solely on sequence information. However, it is important to note that 3D models do not match the efficiency of 2D models in these scenarios, likely due to their susceptibility to noise as discussed in Section 4.1.

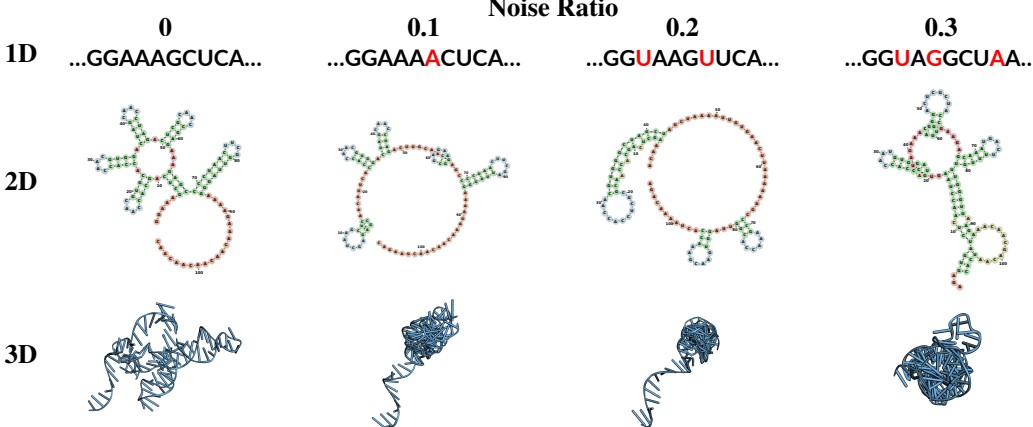

Figure 4: **Visualization of 1D, 2D, and 3D structures under varying noise ratios (mutation errors during sequencing).** Each column represents a different noise ratio, showcasing the impact of noise on the structures across different dimensions.

### 4.3 MODEL ROBUSTNESS AND GENERALIZATION UNDER SEQUENCING NOISE

Tasks 4 and 5 both deal with model performance with noise in the data, but focus on different aspects, robustness to noise, and ability to generalize across unseen noise distributions. As explained in Sec. 3, sequencing noise is common depending on the sequencing method and platform used (Fox et al., 2014), thereby introducing errors in sequences that propagate into 2D and 3D structures. Additionally one of the deployment scenarios involves models trained on high-quality clean data applied for datasets acquired under noisy conditions owing to different sequencing platforms or experimental batch effects (Tom et al., 2017).

To explore these practical aspects, we design two sets of experiments:

- *Robustness:* We introduce sequencing noise into the training, validation, and test sequences to simulate realistic sequencing errors. Nucleotide mutations are applied with probabilities {0.05, 0.1, 0.15, 0.2, 0.25, 0.3}, mirroring typical sequencing error rates (Pfeiffer et al., 2018) which also propagates to the 2D and 3D structures (Fig. 4) via structure prediction tools. Importantly, these mutations are not random; the likelihood of a particular nucleotide mutating into another varies, as

is well documented in sequencing studies Pfeiffer et al. (2018). Our noise model reflects these real-world mutation profiles. Crucially, while the input training, validation, and test sequences contain noise, the property labels remain clean. This again reflects practical scenarios where labels are experimentally determined independent of sequencing and thus unaffected by sequencing errors.

- *Generalization:* Here, models are trained on clean, noise-free data corresponding to high-quality sequencing experiments but are tested on datasets with varying levels of noise simulated by sequencing mutation probabilities in {0.05, 0.1, 0.15, 0.2, 0.25, 0.3}. This setup reflects the real-world scenario where models trained on high-quality data will be deployed on OOD data that may come from different sequencers or have been affected by batch effects.

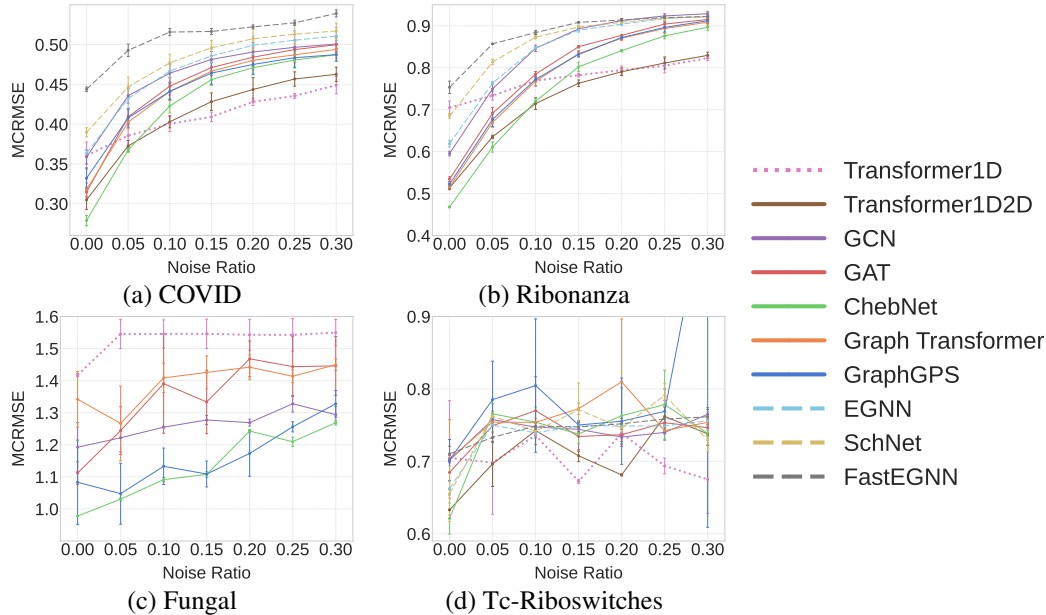

Figure 5: **Robustness experiments.** Transformer1D shows the least performance drop under increasing noise, maintaining the highest accuracy, with Transformer1D2D following closely. In contrast, 2D and 3D models, particularly ChebNet and 3D models, are more impacted by noise.

**Transformer architectures demonstrate superior robustness and generalization under sequencing noise** Expectedly, across both tasks, increased noise levels generally leads to worse test MCRMSE for all models, indicating a decline in prediction performance (Fig. 5, Fig. 6 and Appendix Tables in Sec. I.3, I.4).

Among all models, Transformer1D demonstrates the highest robustness and generalization, exhibiting the least performance degradation as noise levels increase. Notably, in generalization experiments, Transformer1D achieves the best prediction MCRMSE on the COVID, Ribonanza, and Tc-Riboswitches datasets under higher noise levels (Fig. 6 and Appendix I.4). The reliance of Transformer1D on sequence-only information without considering geometric context, while being a weakness in other scenarios, becomes a strength in case of noisy sequences as minor sequencing errors may severely alter the predicted RNA structures (Fig. 4). While its performance on the Fungal dataset is worse overall, Transformer1D maintains remarkably consistent performance as noise increases. Transformer1D2D ranks just behind Transformer1D outperforming other 2D and 3D models. This can be attributed to Transformer1D2D's ability to selectively focus on sequence information for noisy data rather than relying on structural data alone as the self-attention is only weakly conditioned on the graph topology.

More elaborated 2D and 3D models, which rigidly rely on structural information, are significantly more affected by noise, underperforming compared to plain sequence baseline in both robustness and generalization experiments for high noise levels. For low-to-moderate noise levels (5-10% noise), 2D methods such as ChebNet still perform the best. Interestingly, ChebNet shows the worst generalization among 2D models, although it performs on par with other methods in robustness experiments. This suggests that while ChebNet struggles with OOD noise, its performance gets

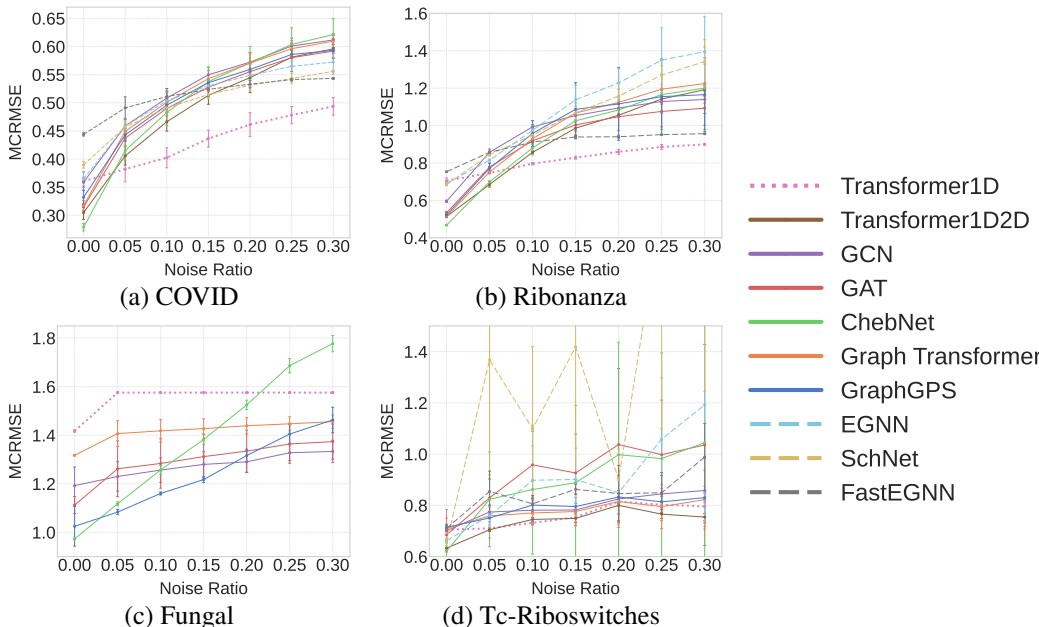

Figure 6: **Generalization experiments.** Transformer1D outperforms other models on noisy sequences, achieving the lowest RMSE at higher noise levels, particularly on COVID, Ribonanza, and Tc-Riboswitches. Transformer1D2D follows closely, showing that transformer-based models generalize better under noise than 2D and 3D models, especially in tasks with geometric representations.

better when it is also trained on the same noise level used during testing, highlighting the need for retraining for different experimental data batches/noise levels in real-world applications. Across both experiments, 3D models have poor performance for more noisy conditions, particularly with the COVID and Ribonanza datasets, due to their dependence on 3D structures which are also sensitive to the propagation of sequencing errors (Fig.4 bottom). In the smaller Tc-Riboswitches dataset, model performances vary more, likely due to limited data size, but transformer models still consistently demonstrate greater robustness to noise.

Across both settings, as the noise ratio rises, the 1D model demonstrates the greatest resilience to noise, showing an average test MCRMSE increase of approximately 14% and 27%, respectively, relative to the train and test on clean data. In contrast, 2D models exhibit the highest sensitivity, with test MCRMSE increasing from 30% to 82%. Meanwhile, 3D models show intermediate performance, with MCRMSE increases of 29% and 56%. Our results reveal a higher vulnerability of 2D and 3D models to sequencing noise where geometric context becomes unreliable at a faster rate than a plain sequence of nucleotides.

## 5 CONCLUSION

We present the first comprehensive study on the benefits and challenges of the effect of geometric context for RNA property prediction models. With providing a curated set of RNA datasets with annotated 2D and 3D structures, we systematically evaluate the performance of 1D, 2D, and 3D models under various real-world conditions, such as limited data, partial labeling, sequencing errors and out-of-distribution generalization. Our results reveal that 2D models outperform 1D and 3D models, with spectral graph neural networks excelling even in low-data and partial labeling scenarios. For 3D models, we find that their potential benefits are hindered by the limited receptive field, computational complexity, and structural noise from RNA structure prediction tools. At the same time, 1D models demonstrate better robustness compared to 2D and 3D models in noisy and OOD conditions. This study highlights the value and limitations of using geometric context for RNA modeling. Future work could focus on ensembling 1D, 2D, and 3D models for complementary strengths, and on improving 2D and 3D models to better handle noise from structure prediction tools as elaborated in Appendix E.2. Another promising direction can be to investigate advanced 3D model architectures which incorporate high-degree steerable features as discussed in Appendix E.1.

ETHICS STATEMENT

This work aims to advance the computational prediction of RNA properties through the development and evaluation of machine learning models utilizing diverse representations of RNA, including 1D sequences, 2D structures, and 3D geometries. In conducting this research, we are committed to upholding high ethical standards in all aspects of our study, ensuring that our work promotes scientific integrity, transparency, and responsible use of technology.

- **Data Integrity and Fair Use:** All RNA datasets used in this study were acquired from publicly available and ethically sourced repositories. Where applicable, the original data sources have been duly cited, and all efforts have been made to ensure data privacy and compliance with relevant data-sharing agreements. Our curated datasets have been handled responsibly, with proper annotations to minimize errors and misinterpretations.

- **Minimization of Bias:** We recognize that RNA data, particularly when it involves limited or incomplete datasets, can introduce biases in model predictions. To mitigate this, we employ a diverse set of RNA data and systematically analyze model performance under various conditions, including label scarcity and sequencing errors. By doing so, we aim to provide balanced insights into the potential advantages and limitations of different modeling approaches, fostering responsible model selection and deployment.

- **Responsible Use of AI in Biology:** The machine learning techniques used in this study are intended to assist in scientific discovery and biological understanding, with potential applications in therapeutic interventions. However, we acknowledge the importance of caution in applying computational models in sensitive domains such as healthcare. While our models are designed to improve RNA property prediction, we emphasize that these predictions should not be used in isolation for clinical or therapeutic decision-making without further validation and consideration of ethical implications.

REPRODUCIBILITY STATEMENT

We are committed to the transparency and reproducibility of our findings. All methodologies, datasets, and benchmarking environments will be made publicly available to the research community, allowing others to reproduce, verify, and extend our work. This open approach aims to advance the field of RNA research and promote collaborative progress. To ensure reproducibility, we also provide detailed descriptions of model architecture and training procedures in the main text. The appendix H contains full details on data preprocessing, hyperparameter searching and additional experimental information. All data and code will be provided to upon acceptance.

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

APPENDIX OVERVIEW

- **Appendix A**: Related Work.
- **Appendix B**: Models Overview.
- **Appendix C**: Memory and Computational Constraints.
- **Appendix D**: Detailed task descriptions.
- **Appendix E**: Discussion on Advanced Model Architectures.
- **Appendix F**: Additional Experimental Information.
- **Appendix G**: Analysis of Noise in 3D Structures.
- **Appendix H**: Reproduction.
- **Appendix I**: Additional Results.

## A  RELATED WORK

**RNA property prediction**  RNA-specific models remain scarce, likely due to limited specialized datasets. Recent advancements in sequence modeling have shown promise, particularly with foundation models like RNA-FM (Chen et al., 2022), UTRBERT (Yang et al., 2023), and RINALMO (Penić et al., 2024), which use transformer-based architectures pre-trained on large RNA sequence corpora to predict various RNA functions and structures. While RNNs and CNNs have been applied to tasks like RNA methylation and protein binding (Wang et al., 2022), they struggle with long-range dependencies (Moskalev et al., 2024). Hybrid models like RNAdegformer (He et al., 2023), combining convolutional layers with self-attention, improve predictions by capturing both local and global dependencies. Although some efforts integrate 2D structures with transformers, explicit 2D and 3D geometric modeling for RNA remains underexplored, with graph-based models mainly focusing on RNA-protein and RNA-drug interaction tasks rather than property prediction (Krishnan et al., 2024; Yan et al., 2020; Arora & Sanguinetti, 2022; Zheng et al., 2022).

**RNA structure prediction**  RNA 2D structure prediction has progressed from dynamic programming methods like Vienna RNAfold (Hofacker et al., 1994) to deep learning-based tools like SPOT-RNA2 (Singh et al., 2021) and UFold (Fu et al., 2022), which enhance accuracy by using neural networks and evolutionary data. Models such as E2Efold (Chen et al., 2020) and RNA-FM (Chen et al., 2022) employ transformer architectures to achieve state-of-the-art results in secondary structure prediction.

RNA 3D structure prediction has progressed through ab initio, template-based, and deep learning approaches. Ab initio methods (e.g., iFoldRNA (Sharma et al., 2008), SimRNA (Boniecki et al., 2016)) balance detail and efficiency but struggle with non-canonical interactions. Template-based models (e.g., FARNA/FARFAR (Das & Baker, 2007), 3dRNA (Zhang et al., 2020)) depend on existing structures but are limited by available data. Deep learning models like DeepFoldRNA (Pearce et al., 2022), RhoFold (Shen et al., 2022), RoseTTAFoldNA (Baek et al., 2024), and trRosettaRNA (Wang et al., 2023) show promise in predicting 3D structures from sequence data but face challenges with novel RNA families due to RNA's conformational flexibility (Kulkarni et al., 2023).

Despite these advances, there is a gap in applying 2D and 3D modeling techniques to RNA property prediction. Most works focus on 1D representations and overlook the potential of geometric information from 2D and 3D structures. This study is the first to systematically explore the benefits and limitations of incorporating explicit structural data in deep learning-based RNA property prediction.

## B  MODELS OVERVIEW

1. **Transformer1D**: The Transformer1D model is a standard Transformer architecture adopted for RNA sequence processing. It includes an embedding layer to convert input tokens into dense vectors, positional encoding (PE) to retain sequence order and a multi-layer Transformer encoder to capture long-range dependencies within the sequence.

2. **Transformer1D2D**: An adaptation of Transformer1D that integrates sequence and 2D graph structure information. The model encodes each nucleotide and incorporates BPP features, combining standard Transformer with positional encoding and a convolutional layer on the graph adjacency matrix. This convolutional output is added to the attention matrix, enabling the model to capture both sequential and structural dependencies.

3. **Graph Convolutional Network (GCN)**: A basic model in graph learning that aggregates and processes node features from local neighborhoods to capture both node characteristics and graph structure, making it effective for tasks like node classification.

4. **Graph Attention Network (GAT)**: Enhances graph convolutions by assigning different importance to neighboring nodes through local attention mechanisms, allowing the model to focus on more relevant nodes during feature aggregation.

5. **ChebNet**: A spectral GNN that utilizes Chebyshev polynomials to approximate the graph Laplacian, enabling graph convolutions with global structural context. This approach allows ChebNet to approximate global graph features with lower computational complexity.

6. **Graph Transformer**: This model uses Laplacian positional encoding to integrate structural information from the graph's Laplacian into node features, which are then processed by Transformer layers. This enables aligning the sequential nature of transformer layers with graph topology.

7. **GraphGPS**: A hybrid model combining GNNs with transformers to capture both local and global graph information. It uses GNNs for local feature aggregation and transformers for long-range dependencies, making it effective for complex graph tasks requiring both local and global context.

8. **SchNet**: An SE(3)-invariant network designed for molecular property prediction on geometric graphs of atomic systems. It operates by modeling interactions through continuous-filter convolutional layers. Since the continuous filter in Schnet is conditioned on distance features, it maintains invariance to rotations and translations of atom coordinates.

9. **E(n)-Equivariant Graph Neural Network (EGNN)**: An equivariant network for geometric graphs with rotations, translations, and reflections symmetry. The EGNN operates as a non-linear message passing between scalar-invariant and vector equivariant quantities.

10. **FAENet:** FAENet, or Frame Averaging Equivariant Network, is a lightweight and scalable Graph Neural Network designed for materials modeling. It introduces Stochastic Frame Averaging to achieve E(3)-equivariance through data transformations rather than architectural constraints, enabling superior accuracy and scalability on tasks like molecular and materials property prediction.

11. **GVPGNN**: Geometric Vector Perceptron GNN is a graph-based model designed to learn from 3D molecular structures by integrating geometric and relational information. It uses Geometric Vector Perceptrons to encode scalar and vector features, enabling effective modeling of both protein structure and interactions.

12. **DimeNet**: Directional Message Passing Neural Network leverages directional message passing to encode both distance and angular information between atoms. It utilizes spherical harmonics and Bessel functions for rotation equivariant message representations, enabling precise modeling of molecular interactions.

13. **FastEGNN**: FastEGNN introduces virtual nodes to traditional equivariant GNNs, enabling global message passing while maintaining scalability. These virtual nodes are designed to approximate and enhance real node interactions, ensuring E(3)-equivariance for efficient modeling of large geometric graphs.

## C  MEMORY AND COMPUTATIONAL CONSTRAINTS

In this section, we compare the models based on run times and GPU memory. Both Transformer1D2D and 3D models (even with nucleotide pooling) encounter out-of-memory (OOM) issues when processing longer sequences, such as those in the Fungal dataset (Table 1). This highlights the need for optimization to handle longer sequences. Figure 7 shows that Transformer1D scales poorly in both runtime and memory due to its expensive attention mechanism, and Transformer1D2D faces additional challenges by processing the sequence and adjacency matrix simultaneously. In contrast, simpler 2D models like GCN, GAT, and ChebNet are more efficient. 3D models

also scale poorly with sequence length due to the increasing number of atoms. Overall, 2D models provide a good balance between computational demands and performance for encoding structural information.

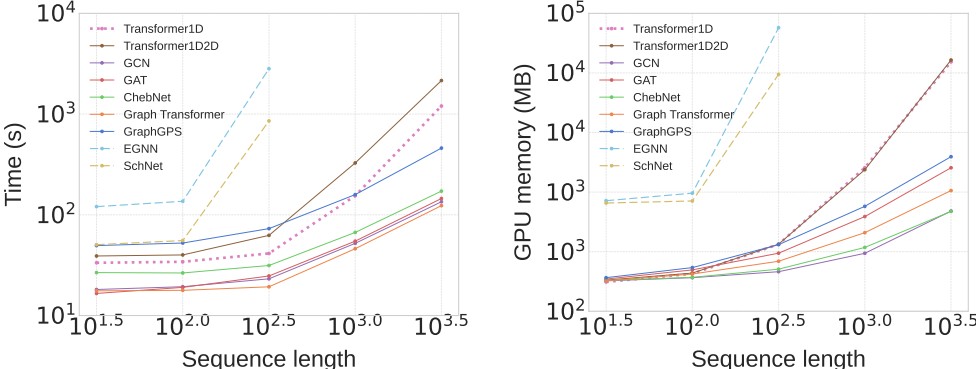

Figure 7: **Running time and memory usage comparison across models.** (1) Running time vs. sequence length (left): Transformer1D and Transformer1D2D scale poorly with sequence length due to expensive attention mechanisms and simultaneous processing of sequences and adjacency matrices, while simpler 2D models like GCN, GAT, and ChebNet are more efficient. (2) Memory usage vs. sequence length (right): Transformer1D2D and 3D models face out-of-memory issues with longer sequences, especially in the largest Fungal dataset, whereas 2D models use memory more efficiently, balancing computational demands and structural encoding.

## D    DETAILED TASK DESCRIPTIONS

To comprehensively evaluate RNA property prediction models, we define five tasks that simulate real-world experimental challenges. These tasks are designed to assess the performance, efficiency, and robustness of models under various conditions encountered in RNA research and application. Below, we provide detailed explanations for each task, their scientific motivation, and their relevance to practical RNA property prediction.

### D.1    TASK 1: IMPACT OF STRUCTURAL INFORMATION ON PREDICTION PERFORMANCE

**Motivation:** RNA molecules fold into complex secondary (2D) and tertiary (3D) structures that dictate their functional properties. While traditional ML approaches often rely solely on the nucleotide sequence (1D representation), it is well-known in chemistry and biology that structural information influences RNA properties (Schlick & Pyle, 2017). However, the extent to which geometric representations can improve performance of ML models remains unclear.

**Task Definition:** We evaluate multiple models across three classes (models described in main text):

- **1D models:** Utilize only the linear sequence of nucleotides.

- **2D models:** Incorporate RNA secondary structure, such as base-pairing interactions.

- **3D models:** Leverage full tertiary structural information, including spatial arrangements of nucleotides.

By comparing their performance across diverse datasets, this task quantifies the benefits of using geometric information. Specifically, we aim to determine if models with explicit 2D or 3D inputs outperform sequence-only models, thereby highlighting the added value of structural representations in RNA property prediction.

**Significance:** This analysis addresses the gap in existing literature, where the impact of structural information on RNA property prediction remains underexplored. The findings inform whether additional computational costs associated with structural modeling are justified by performance gains.

## D.2 TASK 2: MODEL EFFICIENCY IN LIMITED TRAINING DATA SETTINGS

**Motivation:** High-quality RNA datasets with experimentally measured properties are scarce due to the technical difficulty and cost of experimental data acquisition (Byron et al., 2016; Teufel & Sobetzko, 2022). In such scenarios, models must be sample-efficient, learning effectively from small datasets.

**Task Definition:** Let $D = \{X, Y\}$ represent a dataset, where $X$ is the RNA input and $Y$ is the property label. For this task, we define subsets $D_\alpha$ by sampling a fraction $\alpha$ of the original dataset. We evaluate models trained on varying $\alpha$ values (e.g., $\alpha = 25\%, 50\%, 75\%$) to assess how performance scales with reduced training data.

**Significance:** This task simulates real-world scenarios where large labeled datasets are unavailable. It evaluates the ability of different models to generalize from limited data, providing insights into their efficiency and practical applicability.

## D.3 TASK 3: PERFORMANCE WITH PARTIAL SEQUENCE LABELING

**Motivation:** In nucleotide-level RNA datasets, labels for molecular properties (e.g., binding affinities or reactivity) are often available only for a subset of nucleotides in a sequence (Wayment-Steele et al., 2022b). This is due to the high cost of annotating every nucleotide experimentally.

**Task Definition:** We consider datasets where only the first $\ell$-nucleotide positions in each sequence are labeled, representing partial sequence labeling. Models are trained using these incomplete labels and evaluated on their ability to predict properties across the full sequence. Performance is assessed using datasets with varying $\ell$-label fractions.

**Significance:** This task reflects a critical real-world challenge where complete annotations are unavailable. It assesses a model's ability to generalize from partial labels to unseen portions of RNA sequences, a valuable capability for applications with sparse experimental data.

## D.4 TASK 4: ROBUSTNESS TO SEQUENCING NOISE

**Motivation:** RNA sequencing technologies often introduce noise in the form of random nucleotide errors (e.g., insertions, deletions, or substitutions) (Ozsolak & Milos, 2011; Fox et al., 2014). This noise propagates through derived RNA structures (2D and 3D), potentially degrading model performance. Understanding how well models handle noise is crucial for their reliable deployment.

**Task Definition:** To simulate realistic sequencing errors, we introduce controlled noise levels into RNA sequences during both training and testing. Models are evaluated for their ability to maintain performance under consistent noise conditions, representing a scenario where noise characteristics are stable across experimental phases.

**Significance:** This task assesses a model's robustness in practical settings where sequencing noise is unavoidable. It provides insights into the resilience of RNA models to variations in input quality.

## D.5 TASK 5: GENERALIZATION TO OUT-OF-DISTRIBUTION (OOD) DATA

**Motivation:** RNA models are often trained on high-quality experimental datasets but deployed in conditions where data characteristics differ significantly due to variations in sequencing platforms or protocols (Tran et al., 2020; Tom et al., 2017). This mismatch can lead to performance degradation.

**Task Definition:** We evaluate models trained on clean RNA data and tested on datasets with higher levels of noise (representing OOD conditions). Performance metrics are analyzed as a function of increasing noise, quantifying the models' ability to generalize to unseen distributions.

**Significance:** This task mirrors real-world deployment scenarios where models encounter noisy or biased data. It highlights the limitations of models trained on idealized datasets and informs strategies for improving generalization under OOD conditions.

# E    DISCUSSION ON ADVANCED MODEL ARCHITECTURES

## E.1    ENHANCING 3D RNA PREDICTION WITH HIGH-DEGREE STEERABLE FEATURES

Apart from the 3D models considered in this work, models using high-degree steerable features such as TFN (Thomas et al., 2018), MACE (Batatia et al., 2022) and NequIP (Batzner et al., 2022) represent an important aspect of equivariant models, with potential to enhance model expressiveness by incorporating higher-order information. While memory constraints have limited their practical implementation in our current work, techniques such as scalarization of high-degree steerable features, as demonstrated in works such as HEGNN (Frank et al., 2024) and SO3krates (Cen et al., 2024), could address this challenge. For instance, HEGNN can improve the expressivity of equivariant GNNs by mitigating expressivity degeneration on symmetric graphs and leveraging higher-order representations. Although 3D RNA graphs are too complex to be considered symmetric, the ability to handle high-degree steerable features in large RNA graphs, with hundreds of nucleotides and thousands of atoms, remains valuable and has the potential to improve performance. Future work can explore such methods to improve 3D RNA prediction.

## E.2    MITIGATING SEQUENCING NOISE

Notably, most studies in this field do not explicitly address sequencing noise common for RNA in model architecture design, prompting a need to explore effective strategies for mitigating its impact. An effective way to handle noise can be through ensemble methods. For instance, combining 1D, 2D, and 3D models by independently learning representations and integrating them via attention mechanisms that dynamically weigh each modality based on task relevance and noise can leverage the robustness of 1D methods while benefiting from the strengths of 2D and 3D approaches and be a good direction for future research.

# F    ADDITIONAL EXPERIMENTAL INFORMATION

## F.1    DATASET STATISTICS

Here, we present the statistics for each dataset used in the paper in Table 2. The datasets are categorized as small, medium, or large based on the number of sequences and sequence length. "Target" refers to the task the dataset is designed to predict, and "# Avg. Atoms" indicates the average number of atoms used in 3D models.

Table 2: The statistics of Tc-Riboswitches, Ribonanza, COVID, and Fungal datasets.

|  | Tc-Riboswitches | Ribonanza | COVID | Fungal |
| --- | --- | --- | --- | --- |
| Dataset Size | Small | Medium | Medium | Large |
| Task Level | RNA-level | Nucleotide-level | Nucleotide-level | RNA-level |
| Target | Switching Factor | Degradation | Degradation | Expression |
| # Sequences | 355 | 2260 | 4082 | 7089 |
| Sequence Length | 66 - 75 | 177 | 107 - 130 | 150 - 3063 |
| # Labels | 1 | 2 | 3 | 1 |
| # Avg. Atoms (for 3D models) | 1531 | 3791 | 2598 | N/A |

## F.2    COMPARISON OF PARTIAL TRAINING DATA AND PARTIAL SEQUENCE LABELING

To clarify the differences and provide a more detailed explanation, we illustrate two experiments: partial training data and partial sequence labeling (Figure 8).

## F.3    DETAILS OF NOISY EXPERIMENTS: ROBUSTNESS AND GENERALIZATION

To create the noisy datasets, we vary the noise ratio $r$ across the values $\{0.05, 0.1, 0.15, 0.2, 0.25, 0.3\}$. For each given noise ratio, we independently mutate the nucleotide at each position in a

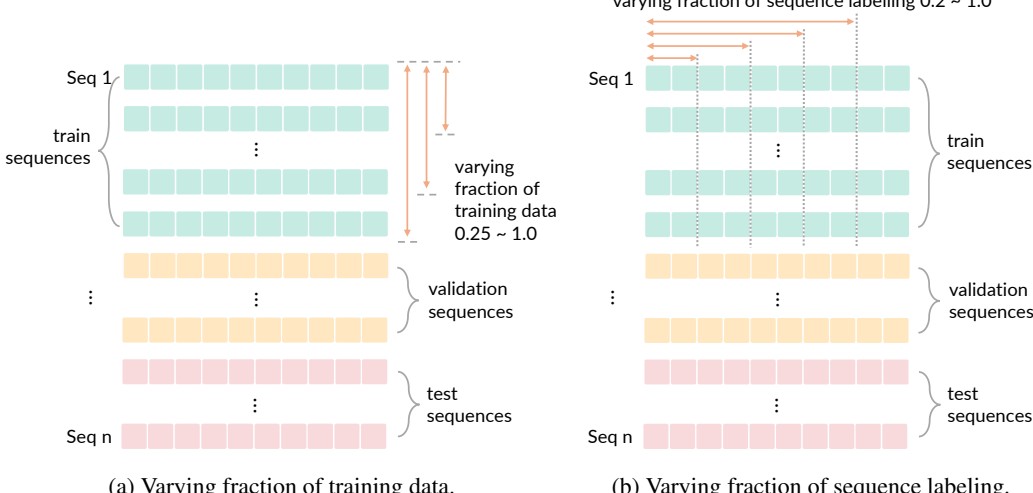

(a) Varying fraction of training data.    (b) Varying fraction of sequence labeling.

Figure 8: **Comparison of partial training data and partial sequence labeling.** The orange arrows indicate the varying components. (a) utilizes full nucleotide labels but trains on varying fractions of RNA sequences (0.25, 0.5, 0.75, 1.0). (b) uses all training sequences but with varying fractions of nucleotide labels (0.2, 0.4, 0.6, 0.8, 1.0). Therefore, (b) is only for nucleotide-level tasks.

sequence with probability $r$, as illustrated in Figure 9. The resulting mutated sequence is then passed to the 2D and 3D prediction tools to generate the corresponding structures. Figure 4 gives a comprehensive illustration of getting noisy 1D, 2D, and 3D structures.

For the robustness experiments, all training, validation, and testing are conducted on the noisy datasets. In contrast, for the generalization experiments, the model is trained and validated on clean datasets, and its performance is tested on noisy datasets with varying noise ratios.

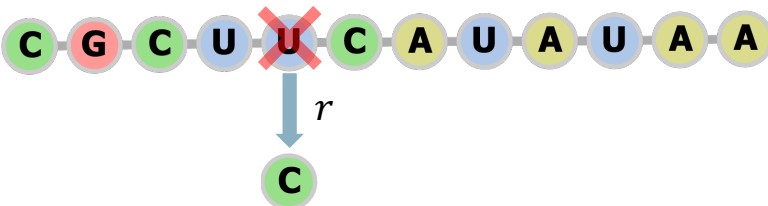

Figure 9: Mutation the nucleotide at each position independently with the probability $r$.

## F.4 ANALYSIS OF TRANSFORMER1D AND TRANSFORMER1D2D

To further validate that incorporating structural information contributes to the final results, we analyze the attention maps generated by Transformer1D and Transformer1D2D. Fig. 10 illustrates the average attention maps across all heads before the final output layer for both the models for a randomly selected RNA sequence. The attention maps of Transformer1D2D exhibit a striking similarity to both the adjacency matrix and the BPP matrix, whereas the attention maps from the standard Transformer model seem to suggest that the model does not learn to attend to the structural features. Moreover, we quantify this observation by computing the cosine similarity between the attention maps of the models and the true adjacency matrix and BPP for all sequences in the COVID dataset. The results, reported in Table 3 show that Transformer1D2D achieves much higher similarity scores compared to the 1D Transformer alone. This reinforces the conclusion that explicit encoding of structural information is essential for improved model performance.

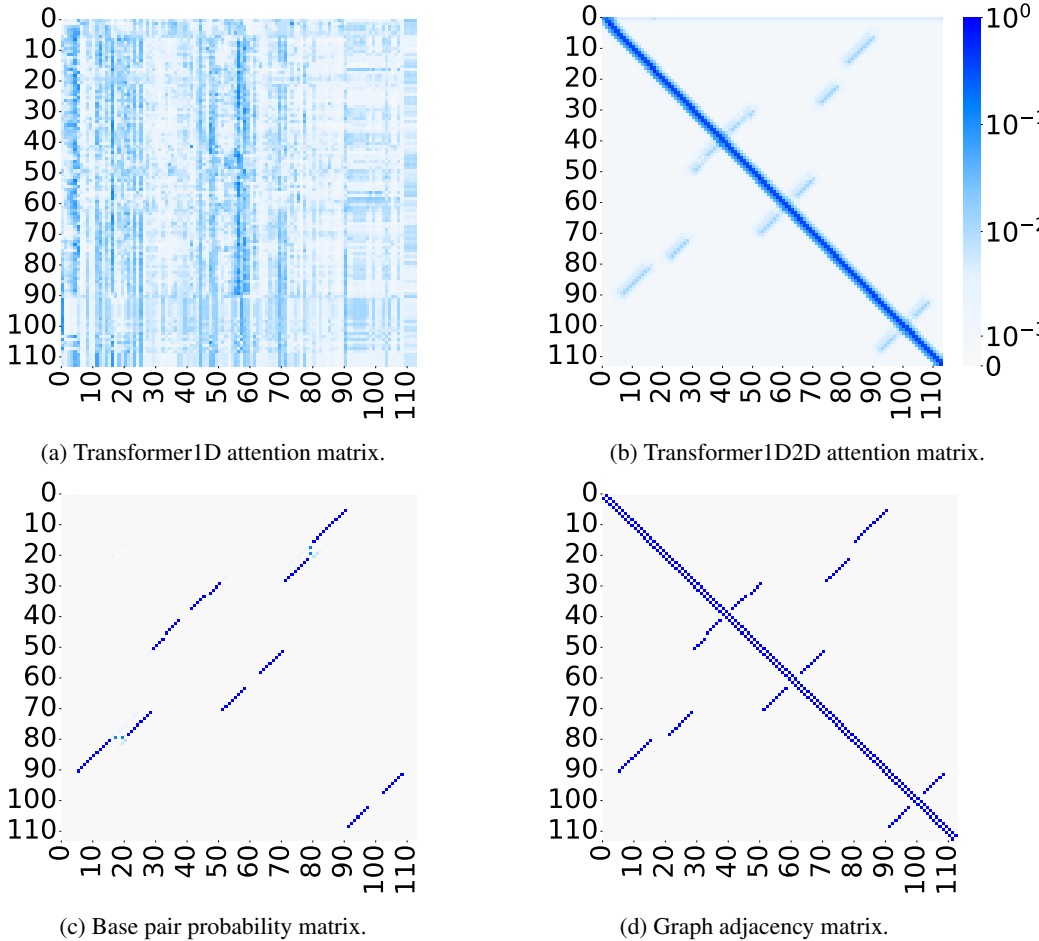

(a) Transformer1D attention matrix.

(b) Transformer1D2D attention matrix.

(c) Base pair probability matrix.

(d) Graph adjacency matrix.

Figure 10: **The heatmaps of matrices in Transformer1D and Transformer1D2D.** Attention maps from Transformer1D2D exhibit a striking resemblance to both the adjacency matrix and BPP matrix, highlighting the model's ability to learn structural features. In contrast, the standard Transformer struggles with this task, as shown by lower cosine similarity scores, reinforcing the conclusion that explicitly encoding structural information is crucial for enhanced model performance.

Table 3: **Cosine similarity values for different models.** Cosine similarity scores between the attention maps and the true adjacency and BPP matrices for all sequences in the COVID dataset demonstrate that Transformer1D2D significantly outperforms the standard Transformer. These results underscore the importance of explicitly encoding structural information for superior model performance.

| Model | Cosine similarity adjacency | Cosine similarity BPP |
|---|---|---|
| Transformer1D | 0.107 | 0.090 |
| Transformer1D2D | 0.448 | 0.672 |

## G    ANALYSIS OF NOISE IN 3D STRUCTURES

As mentioned in the main context, predicted 3D structures consistently exhibit noise. In this section, we analyze this issue from two perspectives: sensitivity to sequence length and variability across different prediction tools.

### G.1 Impact of Sequence Length on 3D Structure Prediction Noise

To investigate the hypothesis that longer sequences result in greater noise in 3D structure predictions, we randomly selected a COVID and Tc-Riboswitches dataset sequence and generated structures using four state-of-the-art 3D structure prediction tools: RhoFold (Shen et al., 2022), RNAComposer (Xu et al., 2014), trRosetta (Baek et al., 2024), and SimRNA (Boniecki et al., 2016). High variability among these predicted structures would indicate significant uncertainty in absolute atom positions. We quantified this noise by aligning the structures using the Kabsch algorithm (Kabsch, 1976) and computing the pairwise RMSD values, resulting in a 4x4 matrix showing structural deviations between each pair of tools (see Table 4). The observed pairwise RMSD values ranged from 16 to 45 Å for the COVID dataset and from 11 to 15 Å for the Tc-Riboswitches dataset, reflecting substantial variability and suggesting considerable noise in the 3D predictions. This level of structural inaccuracy likely contributes to the poorer performance of 3D models. However, we found that 3D models outperform 1D models for shorter sequences, such as those in the Tc-Riboswitches dataset (67 to 73 nucleotides long). This improved performance is due to the lower noise in 3D predictions for shorter sequences, a phenomenon supported by previous studies (Nithin et al., 2024; Ponce-Salvatierra et al., 2019) and also exhibited by the comparatively smaller RMSD values reported in Table 4 for Tc-Riboswitches dataset. The reduced complexity of shorter sequences allows 3D models to capture structural details more accurately, thereby enhancing performance and validating that accurate 3D structure encoding can outperform 1D models.

Table 4: **Pairwise RMSD values (in Å) between 3D structure prediction tools for the COVID dataset (left) and Tc-Riboswitches dataset (right).** The results indicate larger noise in predictions for longer COVID sequences.

| COVID | RhoFold | trRosetta | SimRNA | Composer |
|---|---|---|---|---|
| **RhoFold** | 0 | 39.05192 | 44.76146 | 45.45994 |
| **trRosetta** | 39.05192 | 0 | 22.54974 | 18.17359 |
| **SimRNA** | 44.76146 | 22.54974 | 0 | 16.73399 |
| **Composer** | 45.45994 | 18.17359 | 16.73399 | 0 |

| Tc-Ribo | RhoFold | trRosetta | SimRNA | Composer |
|---|---|---|---|---|
| **RhoFold** | 0 | 14.338 | 11.996 | 15.056 |
| **trRosetta** | 14.338 | 0 | 12.932 | 14.916 |
| **SimRNA** | 11.996 | 12.932 | 0 | 14.243 |
| **Composer** | 15.056 | 14.916 | 14.243 | 0 |

### G.2 Impact of different 3D prediction tools

In this section, we demonstrate the significant differences in 3D structures predicted by various tools. We compare the 3D structure obtained from RhoFold (Shen et al., 2022), which serves as our default method, with those predicted by RNAComposer (Xu et al., 2014), trRosetta (Baek et al., 2024), and SimRNA (Boniecki et al., 2016). Each structure is visualized side by side with RhoFold in Figure 11 to facilitate a more intuitive comparison. As observed, these structures predicted by each tool vary considerably.

## H Reproduction

This section outlines the necessary details to reproduce all experiments discussed in this paper. The code will be made publicly available upon acceptance.

### H.1 Training details

All experiments were conducted on a single NVIDIA A100 GPU. For each baseline, hyperparameters were optimized using Optuna (Akiba et al., 2019), restricting the search to models with fewer

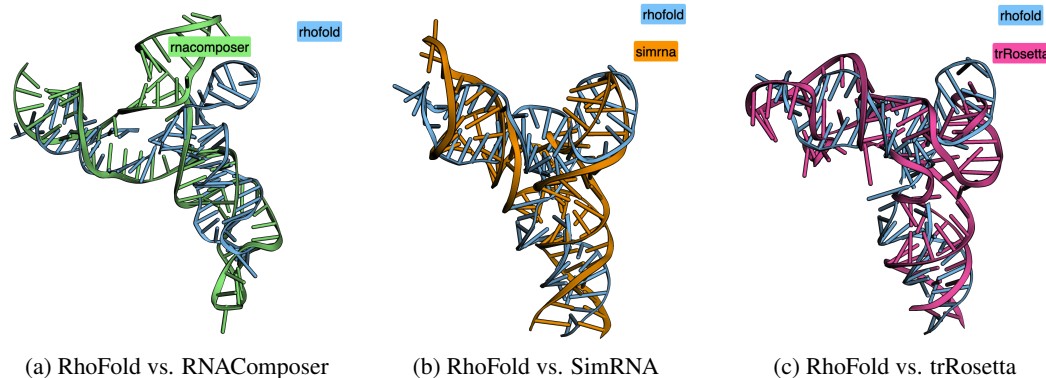

(a) RhoFold vs. RNAComposer      (b) RhoFold vs. SimRNA      (c) RhoFold vs. trRosetta

Figure 11: Comparison of RhoFold against other 3D structure prediction tools on an example sequence from Tc-Riboswitches dataset.

than 10 million parameters that fit within the memory constraints of an 80GB NVIDIA A100 GPU. Most baseline implementations were sourced from PyTorch Geometric (Fey & Lenssen, 2019). The Transformer1D model was adapted to Transformer1D2D as detailed in the paper. For EGNN, we utilized the authors' implementation (Satorras et al., 2021), and for SchNet, the implementation from (Joshi et al., 2023) was used.

## H.2 HYPERPARAMETERS

This section provides a comprehensive overview of the hyperparameters used in each baseline model, facilitating reproducibility and understanding of the model configurations.

Common hyperparameters across these models include `in_channels`, which specifies the number of input features, `hidden`, which determines the number of hidden units in each hidden layer, and `out_channels`, which defines the number of output features. The `L` parameter controls the number of layers in the network, and the `dropout` parameter sets the dropout rate for regularization. The `lr` parameter specifies the learning rate, and `weight_decay` sets the weight decay for regularization of the optimizer. For graph-level tasks, the `pool` parameter specifies the pooling method, which can be `mean`, `max`, or `add`.

**Transformer1D** is a standard Transformer architecture for RNA sequence processing. It includes an embedding layer to convert input tokens into dense vectors, positional encoding (PE) to retain sequence order and a multi-layer Transformer encoder to capture complex dependencies within the sequence. There are some hyperparameters from the original transformer (Vaswani et al., 2017). `nhead`, which defines the number of attention heads in each Transformer layer; `num_encoder_layers`, which controls the number of encoder layers in the Transformer; `d_model`, which determines the dimensionality of the embeddings and the model; `dim_feedforward`, which sets the dimensionality of the feedforward network model. To shrink the search space, we set `d_model` and `dim_feedforward` as the same with a new hyperparameter `hidden`.

**Transformer1D2D** is an adaptation of Transformer1D that integrates both sequence and 2D graph structure information. In addition to encoding each nucleotide, the model incorporates base pair probabilities (BPP) features for each nucleotide. It combines a standard Transformer with positional encoding and a convolutional layer applied to the graph adjacency matrix. This convolutional output is added to the Transformer's attention matrix, allowing the model to incorporate graph structure into its attention mechanism. This design captures both the sequential and structural dependencies in RNA data, improving predictive performance. The unique hyperparameter for this model is `kernel_size`, which specifies the size of the convolutional kernel.

**GAT** includes the unique hyperparameters `gat_heads`, which specify the number of attention heads in each GAT layer.

**ChebNet**   model has the unique hyperparameter `power`, which specifies the polynomial order for the Chebyshev convolution.

**GraphGPS**   and **Graph Transformer** includes `heads`, which specifies the number of attention heads in each layer, and `pe_dim`, which defines the dimensionality of positional encoding.

**EGNN**   and **SchNet** are 3D models that operate at two granularities within the network: atom layers and nucleotide layers. The two types of layers are connected through nucleotide pooling. Atom layers use atoms as nodes, while nucleotide layers use nucleotides as nodes. Both the atom layer and nucleotide layer employ a point cloud setting and calculate edges based on the distance between two nodes. An edge is considered to exist if the distance is smaller than a certain threshold. Therefore, EGNN and SchNet share the following hyperparameters: `L_atom`, which denotes the number of atom layers; `L_nt`, which specifies the number of nucleotide layers; `threshold_atom`, which is the threshold for edges in atom layers; and `threshold_nt`, which is the threshold for edges in nucleotide layers.
For SchNet, the unique hyperparameters include `num_filters`, which refers to the number of filters used in convolutional layers, and `num_gaussians`, which indicates the number of Gaussian functions used for radial filters. For a more detailed explanation of these hyperparameters, please refer to (Schütt et al., 2017).
To ensure a fair comparison, the best hyperparameter configuration for each method was selected based on validation set performance. We report the mean performance and standard deviation across 5 random splits on the test set. For the COVID and Ribonanza datasets, we performed hyperparameter searching only on the COVID dataset and applied the same configuration to Ribonanza, as the two datasets share similar properties. The optimal hyperparameters are shown in Table 5.

## I   ADDITIONAL RESULTS

In this section, we present the additional results supporting Figures 2, 3, 5, and 6 in main text.

### I.1   IMPACT OF DATA AVAILABILITY

The detailed results of partial training data from Figure 2 are shown in Tables 6, 7, and 8.

### I.2   IMPACT OF PARTIAL LABELING

The detailed results of the partial labeling sequence from Figure 3 are shown in Tables 9 and 10.

### I.3   ROBUSTNESS TO SEQUENCING NOISE

The results of the robustness experiment from Figure 5 are shown in Tables 11, 12, 13, and 14.

### I.4   GENERALIZATION TO OOD DATA

The results of the generalization experiment from Figure 6 are shown in Tables 15, 16, 17, and 18.

| | Hyperparameter | COVID & Ribonanza | Tc-riboswitches | Fungal |
|---|---|---|---|---|
| Transformer1D | lr | 0.001 | 0.0005 | 0.001 |
| | weight_decay | 0 | 0.0005 | 0.0005 |
| | hidden | 128 | 64 | 32 |
| | nhead | 8 | 8 | 8 |
| | num_encoder_layers | 8 | 6 | 6 |
| | pool | / | mean | mean |
| Transformer1D2D | lr | 0.001 | 0.005 | |
| | weight_decay | 0 | 0 | |
| | hidden | 256 | 32 | |
| | nhead | 16 | 4 | OOM |
| | num_encoder_layers | 8 | 4 | |
| | pool | mean | mean | |
| | kernel_size | 5 | 5 | |
| GCN | lr | 0.001 | 0.0001 | 0.0001 |
| | weight_decay | 0 | 0 | 0 |
| | hidden | 1024 | 512 | 512 |
| | L | 7 | 5 | 3 |
| | dropout | 0.3 | 0.1 | 0.7 |
| | pool | / | max | add |
| ChebNet | lr | 0.001 | 0.005 | 0.0001 |
| | weight_decay | 0 | 0.0005 | 0 |
| | hidden | 512 | 256 | 256 |
| | L | 5 | 7 | 5 |
| | dropout | 0.3 | 0.3 | 0.3 |
| | power | 6 | 2 | 2 |
| | pool | / | max | mean |
| GAT | lr | 0.001 | 0.005 | 0.0005 |
| | weight_decay | 0 | 0 | 0.0005 |
| | hidden | 256 | 1024 | 256 |
| | L | 7 | 3 | 7 |
| | dropout | 0.1 | 0.1 | 0.3 |
| | heads | 4 | 2 | 1 |
| | pool | / | add | add |
| Graph Transformer | lr | 0.001 | 0.005 | 0.005 |
| | weight_decay | 0 | 0 | 0.0005 |
| | hidden | 128 | 64 | 256 |
| | L | 7 | 5 | 7 |
| | heads | 4 | 1 | 2 |
| | pool | / | add | mean |
| GraphGPS | lr | 0.001 | 1e-5 | 0.0005 |
| | weight_decay | 0 | 0.0005 | 0.0005 |
| | hidden | 256 | 256 | 512 |
| | L | 5 | 7 | 5 |
| | heads | 2 | 1 | 2 |
| | pool | / | add | max |
| EGNN | lr | 0.0005 | 0.001 | |
| | weight_decay | 0 | 0 | |
| | hidden | 256 | 256 | |
| | L_atom | 3 | 3 | OOM |
| | L_nt | 2 | 1 | |
| | threshold_atom | 1.6 | 1.6 | |
| | threshold_nt | 22 | 22 | |
| SchNet | lr | 0.0005 | 0.001 | |
| | weight_decay | 0 | 0 | |
| | hidden | 128 | 128 | |
| | L_atom | 1 | 1 | |
| | L_nt | 2 | 4 | OOM |
| | threshold_atom | 1.6 | 1.8 | |
| | threshold_nt | 44 | 88 | |
| | num_filters | 128 | 256 | |
| | num_gaussians | 50 | 50 | |

Table 5: Best hyperparameters for each model and dataset. Hyperparameters are searched by Optuna. COVID and Ribonanza share the same hyperparameters.

Table 6: Performance (MCRMSE) of different models across various training data (mean ± standard deviation) fractions on COVID dataset.

| COVID | 0.25 | 0.50 | 0.75 | 1.00 |
|---|---|---|---|---|
| Transformer1D | 0.429 ± 0.018 | 0.410 ± 0.016 | 0.375 ± 0.018 | 0.361 ± 0.017 |
| Transformer1D2D | 0.345 ± 0.010 | 0.330 ± 0.011 | 0.306 ± 0.014 | 0.305 ± 0.012 |
| GCN | 0.389 ± 0.008 | 0.377 ± 0.009 | 0.358 ± 0.014 | 0.359 ± 0.009 |
| GAT | 0.352 ± 0.011 | 0.342 ± 0.009 | 0.320 ± 0.014 | 0.315 ± 0.006 |
| ChebNet | 0.320 ± 0.011 | 0.309 ± 0.009 | 0.286 ± 0.018 | 0.279 ± 0.007 |
| Graph Transformer | 0.356 ± 0.008 | 0.344 ± 0.007 | 0.324 ± 0.016 | 0.318 ± 0.008 |
| GraphGPS | 0.367 ± 0.018 | 0.362 ± 0.005 | 0.344 ± 0.010 | 0.332 ± 0.013 |
| EGNN | 0.398 ± 0.001 | 0.391 ± 0.013 | 0.368 ± 0.013 | 0.364 ± 0.003 |
| SchNet | 0.419 ± 0.003 | 0.414 ± 0.011 | 0.392 ± 0.011 | 0.390 ± 0.006 |
| FastEGNN | 0.460 ± 0.018 | 0.452 ± 0.015 | 0.443 ± 0.016 | 0.444 ± 0.003 |

Table 7: Performance (MCRMSE) of different models across various fractions of training data (mean ± standard deviation) on Ribonanza dataset.

| Ribonanza | 0.25 | 0.50 | 0.75 | 1.00 |
|---|---|---|---|---|
| Transformer1D | 0.777 ± 0.014 | 0.740 ± 0.005 | 0.739 ± 0.001 | 0.705 ± 0.015 |
| Transformer1D2D | 0.630 ± 0.016 | 0.553 ± 0.015 | 0.541 ± 0.018 | 0.514 ± 0.004 |
| GCN | 0.668 ± 0.018 | 0.618 ± 0.017 | 0.612 ± 0.013 | 0.595 ± 0.006 |
| GAT | 0.600 ± 0.018 | 0.553 ± 0.026 | 0.544 ± 0.012 | 0.534 ± 0.006 |
| ChebNet | 0.537 ± 0.019 | 0.494 ± 0.022 | 0.499 ± 0.007 | 0.468 ± 0.002 |
| Graph Transformer | 0.567 ± 0.019 | 0.529 ± 0.013 | 0.529 ± 0.010 | 0.515 ± 0.001 |
| GraphGPS | 0.581 ± 0.021 | 0.529 ± 0.015 | 0.540 ± 0.004 | 0.523 ± 0.003 |
| EGNN | 0.694 ± 0.010 | 0.650 ± 0.010 | 0.632 ± 0.015 | 0.619 ± 0.007 |
| SchNet | 0.768 ± 0.008 | 0.724 ± 0.013 | 0.715 ± 0.015 | 0.685 ± 0.006 |
| FastEGNN | 0.795 ± 0.007 | 0.788 ± 0.002 | 0.774 ± 0.016 | 0.753 ± 0.015 |

Table 8: Performance (MCRMSE) of different models across various fractions of training data (mean ± standard deviation) on Fungal dataset.

| Fungal | 0.25 | 0.50 | 0.75 | 1.00 |
|---|---|---|---|---|
| Transformer1D | 1.510 ± 0.006 | 1.446 ± 0.014 | 1.475 ± 0.035 | 1.417 ± 0.005 |
| GCN | 1.243 ± 0.064 | 1.244 ± 0.128 | 1.151 ± 0.077 | 1.192 ± 0.077 |
| GAT | 1.211 ± 0.125 | 1.168 ± 0.033 | 1.146 ± 0.109 | 1.112 ± 0.035 |
| ChebNet | 1.125 ± 0.097 | 1.011 ± 0.010 | 1.008 ± 0.009 | 0.973 ± 0.003 |
| Graph Transformer | 1.415 ± 0.014 | 1.331 ± 0.163 | 1.306 ± 0.127 | 1.317 ± 0.002 |
| GraphGPS | 1.289 ± 0.071 | 1.377 ± 0.127 | 1.357 ± 0.106 | 1.025 ± 0.081 |

Table 9: Performance (MCRMSE) of different models across various fractions of sequence labeling (mean ± standard deviation) on COVID dataset.

| COVID | 0.2 | 0.4 | 0.6 | 0.8 | 1.0 |
|---|---|---|---|---|---|
| Transformer1D | 0.654 ± 0.040 | 0.559 ± 0.011 | 0.480 ± 0.004 | 0.429 ± 0.034 | 0.361 ± 0.017 |
| Transformer1D2D | 0.502 ± 0.002 | 0.470 ± 0.052 | 0.374 ± 0.007 | 0.325 ± 0.006 | 0.305 ± 0.012 |
| GCN | 0.450 ± 0.012 | 0.416 ± 0.012 | 0.397 ± 0.012 | 0.378 ± 0.011 | 0.359 ± 0.009 |
| GAT | 0.411 ± 0.010 | 0.376 ± 0.012 | 0.360 ± 0.012 | 0.336 ± 0.009 | 0.315 ± 0.006 |
| ChebNet | 0.380 ± 0.007 | 0.344 ± 0.008 | 0.325 ± 0.009 | 0.299 ± 0.007 | 0.279 ± 0.007 |
| Graph Transformer | 0.415 ± 0.012 | 0.379 ± 0.011 | 0.362 ± 0.011 | 0.338 ± 0.004 | 0.318 ± 0.008 |
| GraphGPS | 0.428 ± 0.015 | 0.400 ± 0.017 | 0.376 ± 0.013 | 0.351 ± 0.007 | 0.332 ± 0.013 |
| EGNN | 0.436 ± 0.014 | 0.421 ± 0.010 | 0.407 ± 0.004 | 0.385 ± 0.006 | 0.364 ± 0.003 |
| SchNet | 0.442 ± 0.004 | 0.429 ± 0.005 | 0.413 ± 0.001 | 0.407 ± 0.005 | 0.390 ± 0.006 |
| FastEGNN | 0.497 ± 0.004 | 0.490 ± 0.007 | 0.466 ± 0.007 | 0.469 ± 0.009 | 0.444 ± 0.003 |

Table 10: Performance (MCRMSE) of different models across various fractions of sequence labeling (mean ± standard deviation) on Ribonanza dataset.

| Ribonanza | 0.2 | 0.4 | 0.6 | 0.8 | 1.0 |
|---|---|---|---|---|---|
| Transformer1D | 1.137 ± 0.163 | 0.929 ± 0.023 | 0.823 ± 0.018 | 0.742 ± 0.013 | 0.705 ± 0.015 |
| Transformer1D2D | 0.859 ± 0.025 | 0.638 ± 0.013 | 0.632 ± 0.028 | 0.568 ± 0.013 | 0.514 ± 0.004 |
| GCN | 1.191 ± 0.031 | 1.026 ± 0.079 | 1.111 ± 0.206 | 1.070 ± 0.137 | 0.595 ± 0.006 |
| GAT | 0.703 ± 0.015 | 0.632 ± 0.025 | 0.612 ± 0.030 | 0.560 ± 0.010 | 0.534 ± 0.006 |
| ChebNet | 0.614 ± 0.013 | 0.546 ± 0.008 | 0.540 ± 0.008 | 0.514 ± 0.006 | 0.468 ± 0.002 |
| Graph Transformer | 0.719 ± 0.043 | 0.607 ± 0.020 | 0.584 ± 0.015 | 0.552 ± 0.013 | 0.515 ± 0.001 |
| GraphGPS | 0.743 ± 0.058 | 0.663 ± 0.026 | 0.627 ± 0.024 | 0.651 ± 0.026 | 0.523 ± 0.003 |
| EGNN | 0.882 ± 0.010 | 0.722 ± 0.021 | 0.687 ± 0.008 | 0.665 ± 0.013 | 0.619 ± 0.007 |
| SchNet | 0.810 ± 0.002 | 0.781 ± 0.009 | 0.750 ± 0.009 | 0.725 ± 0.004 | 0.685 ± 0.006 |
| FastEGNN | 1.223 ± 0.008 | 0.929 ± 0.008 | 0.860 ± 0.004 | 0.837 ± 0.004 | 0.753 ± 0.015 |

Table 11: Performance (MCRMSE) of various models in **robustness** experiments on the **COVID** dataset (mean ± standard deviation).

| Model | 0.00 | 0.05 | 0.10 | 0.15 | 0.20 | 0.25 | 0.30 |
|---|---|---|---|---|---|---|---|
| Transformer1D | 0.361 ± 0.017 | 0.386 ± 0.015 | 0.400 ± 0.010 | 0.409 ± 0.006 | 0.428 ± 0.005 | 0.435 ± 0.003 | 0.449 ± 0.011 |
| Transformer1D2D | 0.305 ± 0.012 | 0.373 ± 0.007 | 0.403 ± 0.007 | 0.428 ± 0.011 | 0.444 ± 0.015 | 0.457 ± 0.009 | 0.463 ± 0.009 |
| GCN | 0.359 ± 0.009 | 0.436 ± 0.009 | 0.464 ± 0.011 | 0.481 ± 0.010 | 0.491 ± 0.012 | 0.497 ± 0.009 | 0.501 ± 0.009 |
| GAT | 0.315 ± 0.006 | 0.409 ± 0.009 | 0.448 ± 0.011 | 0.471 ± 0.010 | 0.484 ± 0.012 | 0.494 ± 0.011 | 0.500 ± 0.010 |
| ChebNet | 0.279 ± 0.007 | 0.368 ± 0.003 | 0.423 ± 0.009 | 0.456 ± 0.007 | 0.471 ± 0.009 | 0.481 ± 0.010 | 0.487 ± 0.008 |
| Graph Transformer | 0.318 ± 0.008 | 0.403 ± 0.008 | 0.441 ± 0.012 | 0.467 ± 0.011 | 0.480 ± 0.012 | 0.487 ± 0.010 | 0.494 ± 0.011 |
| GraphGPS | 0.332 ± 0.013 | 0.408 ± 0.012 | 0.441 ± 0.010 | 0.464 ± 0.014 | 0.475 ± 0.012 | 0.484 ± 0.012 | 0.487 ± 0.008 |
| EGNN | 0.364 ± 0.003 | 0.432 ± 0.012 | 0.467 ± 0.009 | 0.486 ± 0.009 | 0.499 ± 0.011 | 0.505 ± 0.012 | 0.511 ± 0.011 |
| SchNet | 0.390 ± 0.006 | 0.447 ± 0.012 | 0.477 ± 0.011 | 0.496 ± 0.009 | 0.507 ± 0.014 | 0.513 ± 0.012 | 0.517 ± 0.010 |
| FastEGNN | 0.444 ± 0.003 | 0.49283 ± 0.008 | 0.516 ± 0.005 | 0.516 ± 0.004 | 0.522 ± 0.002 | 0.527 ± 0.003 | 0.540 ± 0.004 |

Table 12: Performance (MCRMSE) of various models in **robustness** experiments on the **Ribonanza** dataset (mean ± standard deviation).

| Model | 0.00 | 0.05 | 0.10 | 0.15 | 0.20 | 0.25 | 0.30 |
|---|---|---|---|---|---|---|---|
| Transformer1D | 0.705 ± 0.015 | 0.733 ± 0.010 | 0.769 ± 0.014 | 0.782 ± 0.005 | 0.794 ± 0.010 | 0.805 ± 0.017 | 0.823 ± 0.005 |
| Transformer1D2D | 0.514 ± 0.004 | 0.635 ± 0.004 | 0.714 ± 0.014 | 0.763 ± 0.008 | 0.790 ± 0.009 | 0.811 ± 0.014 | 0.830 ± 0.008 |
| GCN | 0.595 ± 0.006 | 0.750 ± 0.014 | 0.846 ± 0.008 | 0.893 ± 0.003 | 0.912 ± 0.005 | 0.924 ± 0.005 | 0.929 ± 0.005 |
| GAT | 0.534 ± 0.006 | 0.691 ± 0.015 | 0.785 ± 0.006 | 0.850 ± 0.003 | 0.877 ± 0.001 | 0.904 ± 0.007 | 0.915 ± 0.007 |
| ChebNet | 0.468 ± 0.002 | 0.611 ± 0.012 | 0.720 ± 0.006 | 0.802 ± 0.011 | 0.841 ± 0.003 | 0.876 ± 0.007 | 0.897 ± 0.008 |
| Graph Transformer | 0.515 ± 0.001 | 0.670 ± 0.011 | 0.768 ± 0.011 | 0.833 ± 0.008 | 0.870 ± 0.006 | 0.893 ± 0.010 | 0.908 ± 0.007 |
| GraphGPS | 0.523 ± 0.003 | 0.677 ± 0.017 | 0.772 ± 0.006 | 0.832 ± 0.006 | 0.872 ± 0.004 | 0.896 ± 0.011 | 0.912 ± 0.006 |
| EGNN | 0.619 ± 0.007 | 0.764 ± 0.003 | 0.847 ± 0.003 | 0.889 ± 0.005 | 0.904 ± 0.003 | 0.917 ± 0.000 | 0.922 ± 0.002 |
| SchNet | 0.685 ± 0.006 | 0.814 ± 0.006 | 0.873 ± 0.004 | 0.897 ± 0.004 | 0.908 ± 0.004 | 0.918 ± 0.005 | 0.922 ± 0.005 |
| FastEGNN | 0.753 ± 0.015 | 0.857 ± 0.001 | 0.884 ± 0.005 | 0.908 ± 0.001 | 0.914 ± 0.004 | 0.920 ± 0.003 | 0.922 ± 0.003 |

Table 13: Performance (MCRMSE) of various models in **robustness** experiments on the **Fungal** dataset (mean ± standard deviation).

| Model | 0.00 | 0.05 | 0.10 | 0.15 | 0.20 | 0.25 | 0.30 |
|---|---|---|---|---|---|---|---|
| Transformer1D | 1.417 ± 0.005 | 1.545 ± 0.045 | 1.546 ± 0.044 | 1.546 ± 0.046 | 1.543 ± 0.048 | 1.543 ± 0.051 | 1.550 ± 0.041 |
| GCN | 1.192 ± 0.077 | 1.222 ± 0.044 | 1.255 ± 0.002 | 1.277 ± 0.014 | 1.269 ± 0.011 | 1.328 ± 0.026 | 1.294 ± 0.025 |
| GAT | 1.112 ± 0.035 | 1.244 ± 0.074 | 1.391 ± 0.155 | 1.334 ± 0.099 | 1.468 ± 0.056 | 1.444 ± 0.094 | 1.446 ± 0.092 |
| ChebNet | 0.978 ± 0.000 | 1.031 ± 0.003 | 1.091 ± 0.007 | 1.108 ± 0.009 | 1.243 ± 0.005 | 1.210 ± 0.014 | 1.269 ± 0.007 |
| Graph Transformer | 1.342 ± 0.087 | 1.267 ± 0.116 | 1.409 ± 0.046 | 1.426 ± 0.051 | 1.442 ± 0.038 | 1.413 ± 0.020 | 1.450 ± 0.018 |
| GraphGPS | 1.083 ± 0.131 | 1.048 ± 0.095 | 1.133 ± 0.057 | 1.109 ± 0.040 | 1.173 ± 0.071 | 1.256 ± 0.016 | 1.328 ± 0.041 |

Table 14: Performance (MCRMSE) of various models in **robustness** experiments on the **Tc-riboswitches** dataset (mean ± standard deviation).

| Model | 0.00 | 0.05 | 0.10 | 0.15 | 0.20 | 0.25 | 0.30 |
|---|---|---|---|---|---|---|---|
| Transformer1D | 0.705 ± 0.079 | 0.698 ± 0.071 | 0.736 ± 0.004 | 0.672 ± 0.003 | 0.739 ± 0.008 | 0.694 ± 0.011 | 0.675 ± 0.047 |
| Transformer1D2D | 0.633 ± 0.001 | 0.697 ± 0.031 | 0.742 ± 0.003 | 0.708 ± 0.008 | 0.681 ± 0.001 | 0.762 ± 0.022 | 0.738 ± 0.016 |
| GCN | 0.701 ± 0.004 | 0.758 ± 0.003 | 0.747 ± 0.005 | 0.744 ± 0.004 | 0.733 ± 0.013 | 0.740 ± 0.011 | 0.765 ± 0.009 |
| GAT | 0.685 ± 0.024 | 0.749 ± 0.017 | 0.770 ± 0.047 | 0.734 ± 0.021 | 0.737 ± 0.009 | 0.753 ± 0.001 | 0.747 ± 0.011 |
| ChebNet | 0.621 ± 0.022 | 0.766 ± 0.014 | 0.754 ± 0.021 | 0.738 ± 0.014 | 0.763 ± 0.039 | 0.778 ± 0.048 | 0.739 ± 0.004 |
| Graph Transformer | 0.703 ± 0.054 | 0.754 ± 0.005 | 0.754 ± 0.006 | 0.773 ± 0.008 | 0.810 ± 0.087 | 0.742 ± 0.004 | 0.754 ± 0.005 |
| GraphGPS | 0.702 ± 0.028 | 0.785 ± 0.053 | 0.805 ± 0.092 | 0.750 ± 0.006 | 0.755 ± 0.060 | 0.769 ± 0.031 | 1.078 ± 0.469 |
| EGNN | 0.663 ± 0.010 | 0.750 ± 0.001 | 0.739 ± 0.002 | 0.749 ± 0.005 | 0.749 ± 0.001 | 0.749 ± 0.001 | 0.756 ± 0.013 |
| SchNet | 0.655 ± 0.038 | 0.762 ± 0.005 | 0.742 ± 0.002 | 0.771 ± 0.037 | 0.746 ± 0.005 | 0.791 ± 0.016 | 0.730 ± 0.016 |
| FastEGNN | 0.710 ± 0.010 | 0.733 ± 0.007 | 0.749 ± 0.006 | 0.748 ± 0.006 | 0.752 ± 0.008 | 0.758 ± 0.017 | 0.761 ± 0.010 |

Table 15: Performance (MCRMSE) of various models in **generalization** experiments on the **COVID** dataset (mean ± standard deviation).

| Model | 0.00 | 0.05 | 0.10 | 0.15 | 0.20 | 0.25 | 0.30 |
|---|---|---|---|---|---|---|---|
| Transformer1D | $0.361 \pm 0.017$ | $0.382 \pm 0.022$ | $0.402 \pm 0.018$ | $0.436 \pm 0.015$ | $0.461 \pm 0.021$ | $0.478 \pm 0.015$ | $0.494 \pm 0.016$ |
| Transformer1D2D | $0.305 \pm 0.012$ | $0.406 \pm 0.016$ | $0.466 \pm 0.017$ | $0.513 \pm 0.016$ | $0.545 \pm 0.027$ | $0.581 \pm 0.025$ | $0.596 \pm 0.018$ |
| GCN | $0.359 \pm 0.009$ | $0.459 \pm 0.011$ | $0.508 \pm 0.011$ | $0.550 \pm 0.014$ | $0.572 \pm 0.016$ | $0.601 \pm 0.014$ | $0.612 \pm 0.008$ |
| GAT | $0.315 \pm 0.006$ | $0.437 \pm 0.013$ | $0.490 \pm 0.013$ | $0.528 \pm 0.008$ | $0.555 \pm 0.013$ | $0.580 \pm 0.015$ | $0.592 \pm 0.012$ |
| ChebNet | $0.279 \pm 0.007$ | $0.415 \pm 0.017$ | $0.483 \pm 0.023$ | $0.538 \pm 0.025$ | $0.571 \pm 0.029$ | $0.604 \pm 0.030$ | $0.621 \pm 0.028$ |
| Graph Transformer | $0.318 \pm 0.008$ | $0.449 \pm 0.015$ | $0.501 \pm 0.018$ | $0.543 \pm 0.015$ | $0.571 \pm 0.019$ | $0.596 \pm 0.014$ | $0.609 \pm 0.014$ |
| GraphGPS | $0.332 \pm 0.013$ | $0.443 \pm 0.011$ | $0.496 \pm 0.006$ | $0.536 \pm 0.005$ | $0.559 \pm 0.010$ | $0.586 \pm 0.007$ | $0.593 \pm 0.005$ |
| EGNN | $0.365 \pm 0.011$ | $0.458 \pm 0.014$ | $0.504 \pm 0.018$ | $0.530 \pm 0.020$ | $0.549 \pm 0.021$ | $0.565 \pm 0.022$ | $0.572 \pm 0.022$ |
| SchNet | $0.390 \pm 0.006$ | $0.457 \pm 0.011$ | $0.491 \pm 0.008$ | $0.515 \pm 0.007$ | $0.531 \pm 0.010$ | $0.543 \pm 0.009$ | $0.556 \pm 0.002$ |
| FastEGNN | $0.444 \pm 0.003$ | $0.491 \pm 0.020$ | $0.511 \pm 0.014$ | $0.524 \pm 0.009$ | $0.533 \pm 0.006$ | $0.541 \pm 0.003$ | $0.543 \pm 0.001$ |

Table 16: Performance (MCRMSE) of various models in **generalization** experiments on the **Ribonanza** dataset (mean ± standard deviation).

| Model | 0.00 | 0.05 | 0.10 | 0.15 | 0.20 | 0.25 | 0.30 |
|---|---|---|---|---|---|---|---|
| Transformer1D | $0.705 \pm 0.015$ | $0.747 \pm 0.005$ | $0.796 \pm 0.006$ | $0.828 \pm 0.008$ | $0.860 \pm 0.013$ | $0.886 \pm 0.013$ | $0.899 \pm 0.003$ |
| Transformer1D2D | $0.514 \pm 0.004$ | $0.685 \pm 0.014$ | $0.857 \pm 0.008$ | $0.986 \pm 0.015$ | $1.055 \pm 0.007$ | $1.142 \pm 0.020$ | $1.192 \pm 0.034$ |
| GCN | $0.595 \pm 0.006$ | $0.857 \pm 0.018$ | $0.993 \pm 0.012$ | $1.054 \pm 0.034$ | $1.094 \pm 0.043$ | $1.129 \pm 0.061$ | $1.139 \pm 0.075$ |
| GAT | $0.534 \pm 0.006$ | $0.778 \pm 0.021$ | $0.919 \pm 0.030$ | $1.003 \pm 0.056$ | $1.047 \pm 0.073$ | $1.076 \pm 0.082$ | $1.093 \pm 0.091$ |
| ChebNet | $0.468 \pm 0.002$ | $0.699 \pm 0.005$ | $0.881 \pm 0.038$ | $1.025 \pm 0.095$ | $1.083 \pm 0.111$ | $1.165 \pm 0.185$ | $1.200 \pm 0.220$ |
| Graph Transformer | $0.515 \pm 0.001$ | $0.752 \pm 0.005$ | $0.930 \pm 0.013$ | $1.067 \pm 0.036$ | $1.124 \pm 0.033$ | $1.194 \pm 0.080$ | $1.224 \pm 0.104$ |
| GraphGPS | $0.523 \pm 0.003$ | $0.771 \pm 0.026$ | $0.958 \pm 0.068$ | $1.087 \pm 0.142$ | $1.116 \pm 0.195$ | $1.154 \pm 0.202$ | $1.165 \pm 0.196$ |
| EGNN | $0.691 \pm 0.006$ | $0.815 \pm 0.004$ | $0.975 \pm 0.026$ | $1.138 \pm 0.078$ | $1.228 \pm 0.079$ | $1.350 \pm 0.173$ | $1.395 \pm 0.187$ |
| SchNet | $0.685 \pm 0.006$ | $0.844 \pm 0.006$ | $0.949 \pm 0.022$ | $1.068 \pm 0.035$ | $1.157 \pm 0.069$ | $1.270 \pm 0.049$ | $1.342 \pm 0.117$ |
| FastEGNN | $0.753 \pm 0.015$ | $0.857 \pm 0.001$ | $0.912 \pm 0.007$ | $0.939 \pm 0.011$ | $0.940 \pm 0.010$ | $0.952 \pm 0.008$ | $0.957 \pm 0.001$ |

Table 17: Performance (MCRMSE) of various models in **generalization** experiments on the **Fungal** dataset (mean ± standard deviation).

| Model | 0.00 | 0.05 | 0.10 | 0.15 | 0.20 | 0.25 | 0.30 |
|---|---|---|---|---|---|---|---|
| Transformer1D | $1.417 \pm 0.005$ | $1.575 \pm 0.002$ | $1.575 \pm 0.002$ | $1.575 \pm 0.002$ | $1.575 \pm 0.002$ | $1.575 \pm 0.002$ | $1.575 \pm 0.002$ |
| GCN | $1.192 \pm 0.077$ | $1.230 \pm 0.061$ | $1.256 \pm 0.050$ | $1.280 \pm 0.052$ | $1.290 \pm 0.044$ | $1.328 \pm 0.032$ | $1.333 \pm 0.046$ |
| GAT | $1.112 \pm 0.035$ | $1.262 \pm 0.115$ | $1.284 \pm 0.102$ | $1.312 \pm 0.092$ | $1.334 \pm 0.087$ | $1.364 \pm 0.080$ | $1.373 \pm 0.072$ |
| ChebNet | $0.973 \pm 0.003$ | $1.118 \pm 0.008$ | $1.257 \pm 0.018$ | $1.382 \pm 0.022$ | $1.525 \pm 0.020$ | $1.686 \pm 0.030$ | $1.777 \pm 0.033$ |
| Graph Transformer | $1.317 \pm 0.002$ | $1.407 \pm 0.053$ | $1.418 \pm 0.046$ | $1.427 \pm 0.040$ | $1.439 \pm 0.033$ | $1.447 \pm 0.029$ | $1.456 \pm 0.028$ |
| GraphGPS | $1.025 \pm 0.081$ | $1.083 \pm 0.011$ | $1.160 \pm 0.006$ | $1.217 \pm 0.012$ | $1.316 \pm 0.026$ | $1.405 \pm 0.040$ | $1.463 \pm 0.052$ |

Table 18: Performance (MCRMSE) of various models in **generalization** experiments on the **Tc-riboswitches** dataset (mean ± standard deviation).

| Model | 0.00 | 0.05 | 0.10 | 0.15 | 0.20 | 0.25 | 0.30 |
|---|---|---|---|---|---|---|---|
| Transformer1D | $0.705 \pm 0.079$ | $0.711 \pm 0.038$ | $0.732 \pm 0.007$ | $0.753 \pm 0.019$ | $0.815 \pm 0.091$ | $0.803 \pm 0.062$ | $0.796 \pm 0.079$ |
| Transformer1D2D | $0.633 \pm 0.001$ | $0.705 \pm 0.007$ | $0.745 \pm 0.008$ | $0.749 \pm 0.017$ | $0.800 \pm 0.034$ | $0.766 \pm 0.017$ | $0.754 \pm 0.014$ |
| GCN | $0.701 \pm 0.004$ | $0.774 \pm 0.026$ | $0.781 \pm 0.051$ | $0.782 \pm 0.062$ | $0.825 \pm 0.093$ | $0.845 \pm 0.072$ | $0.858 \pm 0.126$ |
| GAT | $0.685 \pm 0.024$ | $0.829 \pm 0.074$ | $0.958 \pm 0.131$ | $0.926 \pm 0.152$ | $1.037 \pm 0.297$ | $0.998 \pm 0.213$ | $1.036 \pm 0.392$ |
| ChebNet | $0.621 \pm 0.022$ | $0.824 \pm 0.185$ | $0.862 \pm 0.251$ | $0.888 \pm 0.302$ | $0.997 \pm 0.439$ | $0.983 \pm 0.412$ | $1.045 \pm 0.514$ |
| Graph Transformer | $0.710 \pm 0.041$ | $0.759 \pm 0.035$ | $0.770 \pm 0.047$ | $0.776 \pm 0.054$ | $0.815 \pm 0.101$ | $0.795 \pm 0.085$ | $0.822 \pm 0.115$ |
| GraphGPS | $0.715 \pm 0.012$ | $0.751 \pm 0.016$ | $0.801 \pm 0.023$ | $0.795 \pm 0.014$ | $0.833 \pm 0.025$ | $0.814 \pm 0.022$ | $0.830 \pm 0.018$ |
| EGNN | $0.663 \pm 0.010$ | $0.760 \pm 0.043$ | $0.898 \pm 0.134$ | $0.901 \pm 0.123$ | $0.849 \pm 0.053$ | $1.058 \pm 0.239$ | $1.192 \pm 0.453$ |
| SchNet | $0.655 \pm 0.038$ | $1.368 \pm 0.449$ | $1.099 \pm 0.320$ | $1.419 \pm 0.491$ | $0.900 \pm 0.119$ | $1.957 \pm 0.858$ | $2.025 \pm 0.779$ |
| FastEGNN | $0.710 \pm 0.011$ | $0.854 \pm 0.079$ | $0.806 \pm 0.032$ | $0.863 \pm 0.020$ | $0.846 \pm 0.110$ | $0.849 \pm 0.079$ | $0.988 \pm 0.131$ |

