# OpenReview forum: "Beyond Sequence: Impact of Geometric Context for RNA Property Prediction"
_ICLR.cc/2025/Conference — ICLR 2025 Poster_

### Official Review · Reviewer_HBUE · 2024-10-25

**Soundness:** 3
**Presentation:** 3
**Contribution:** 3
**Rating:** 8
**Confidence:** 4

**Summary:**

This paper presents a systematic evaluation of incorporating explicit geometric information into RNA property prediction, considering not only performance but also real-world challenges such as limited data availability, partial labeling, sequencing noise, and computational efficiency. To this end, authors introduce a newly curated set of RNA datasets with enhanced 2D and 3D structural annotations, providing a resource for model evaluation on RNA data.

**Strengths:**

For the field of AI for Science, high-quality datasets are very important assistants. This article integrates four datasets, which contain a large number of data samples of various types.

**Weaknesses:**

> **W1. Lack of explanation for RNA's uniqueness.**

In section 3, it seems that a task of general sequence molecules is defined, and it is not specifically for RNA. Is there any fundamental difference in methodology between them and other sequenced molecules (such as proteins and DNA), except that the molecular composition may be slightly different?

> **W2. The method used is relatively old.**

There are many new works for 1D sequences and 2D topological graphs, which are not elaborated here. As for 3D geometric graph neural networks, many latest works are not included in the comparison. In fact, the EGNN used has been pointed out by some literature to have form capacity limitations [a], which makes the conclusions drawn in the article unreliable. More and stronger baselines should be added. You can refer to these surveys [b,c,d], and add baselines such as SaVeNet [e], LEFTNET [f], FAENet [g], TFN [h], NequIP [i], MACE [j], EquiformerV2 [k], EPT [l], MEAN [m], dyMEAN [n], etc.

[a] On the expressive power of geometric graph neural networks

[b] A Survey of Geometric Graph Neural Networks: Data Structures, Models and Applications

[c] A Hitchhiker's Guide to Geometric GNNs for 3D Atomic Systems

[d] Artificial Intelligence for Science in Quantum, Atomistic, and Continuum Systems

[e] SaVeNet: A Scalable Vector Network for Enhanced Molecular Representation Learning

[f] A new perspective on building efficient and expressive 3D equivariant graph neural networks

[g] FAENet: Frame Averaging Equivariant GNN for Materials Modeling

[h] Tensor field networks: Rotation- and translation-equivariant neural networks for 3D point clouds

[i] E(3)-equivariant graph neural networks for data-efficient and accurate interatomic potentials

[j] Mace: Higher order equivariant message passing neural networks for fast and accurate force fields

[k] EquiformerV2: Improved Equivariant Transformer for Scaling to Higher-Degree Representations

[l] Equivariant Pretrained Transformer for Unified Geometric Learning on Multi-Domain 3D Molecules

[m] Conditional Antibody Design as 3D Equivariant Graph Translation

[n] End-to-End Full-Atom Antibody Design

**Questions:**

See Weakness.

---

> ### Author Response · Authors · 2024-11-21
> **Author Responses (1/2)**
>
> We thank the reviewer for their feedback and appreciating the value and importance of our newly contributed RNA datasets for advancing the field of RNA property prediction. We address the questions and comments raised by the reviewer point-by-point:
>
> **RW1: Lack of explanation for RNA's uniqueness:**
> We thank the reviewer for raising this point. RNA differs fundamentally from proteins in both the availability of data and the nature of practical challenges in modeling. Unlike proteins, which benefit from extensive high-quality databases such as PDB [1] containing sequences, experimental structures, and experimentally measured property labels in benchmarks such as FLIP [2], RNA datasets are far more limited [4,5]. Existing RNA resources, like RNAsolo [3], provide 3D structures but lack property annotations, leaving a significant gap in benchmark datasets for RNA property prediction. Moreover, RNA is particularly susceptible to sequencing noise due to variability in platforms and quality [6, 7], a challenge less prominent in protein studies where experimental characterization techniques are more advanced. These distinctions mean that while the role of structural information is well-understood for proteins, it remains under-explored for RNA. Thus, the evaluation setups for protein and RNA cannot be compared directly. With regards to DNA, similar limitations hold as for RNA and hence we are not aware of high-quality structural datasets with property labels for DNA either and recent literature only chooses to model DNA sequences alone.
>
>
> &nbsp;
>
> References:
> [1] Protein Data Bank: the single global archive for 3D macromolecular structure data, Nucleic acids research, 2019.
> [2] FLIP: Benchmark tasks in fitness landscape inference for proteins, Dallago et al., NeurIPS 2021
> [3] RNAsolo: a repository of cleaned PDB-derived RNA 3D structures, Adamczyk et al., Bioinformatics 2022
> [4] Translating rna sequencing into clinical diagnostics: opportunities and challenges, Byron et al., Nature Reviews Genetics, 2016
> [5] Reducing costs for dna and rna sequencing by sample pooling using a metagenomic approach, Teufel \& Sobetzko, BMC genomics, 2022.
> [6] Rna sequencing: advances, challenges and opportunities, Ozsolak \& Milos, Nature reviews genetics, 2011.
> [7] Accuracy of next generation sequencing platforms, Fox et al., Next generation, sequencing \& applications, 2014.

---

> ### Author Response · Authors · 2024-11-21
> **Author Responses (2/2)**
>
> **RW2: Evaluation with more recent 3D methods:**.
>
> Following the reviewer's suggestion, we have now added a comparison of four additional 3D models (GVP, DimeNet, and recent FAENet and FastEGNN) in Table 1 and Figs. 2, 3, 5, 6 in the main text (highlighted in blue). However, we still find that all these models show similar performance on RNA property prediction tasks with 3D models failing to outperform 2D models. We further chose FastEGNN for all subsequent experiments and observe the similar trend as reported earlier with EGNN and SchNet.
>
> | Model                     | COVID         | Ribonanza       | Tc-Ribo        | Fungal         |
> |---------------------------|---------------|-----------------|----------------|----------------|
> | **1D model**             |               |                 |                |                |
> | Transformer1D            | 0.361±0.017   | 0.705±0.015     | 0.705±0.019    | 1.417±0.005    |
> | **2D model**             |               |                 |                |                |
> | Transformer1D2D          | 0.305±0.012   | 0.514±0.004     | 0.633±0.001    | OOM            |
> | GCN                      | 0.359±0.009   | 0.509±0.004     | 0.640±0.005    | 1.192±0.077    |
> | GAT                      | 0.315±0.006   | 0.534±0.006     | 0.603±0.004    | 1.112±0.035    |
> | ChebNet                  | 0.279±0.015   | 0.499±0.005     | 0.599±0.001    | 1.018±0.023    |
> | Graph Transformer        | 0.318±0.008   | 0.500±0.005     | 0.604±0.001    | 1.317±0.002    |
> | GraphGPS                 | 0.332±0.013   | 0.523±0.003     | 0.610±0.012    | 1.025±0.081    |
> | **3D model (w/o pooling)** |               |                 |                |                |
> | EGNN (w/o pooling)       | 0.480±0.025   | 0.808±0.023     | 0.725±0.002    | OOM            |
> | SchNet (w/o pooling)     | 0.499±0.030   | 0.843±0.004     | 0.704±0.001    | OOM            |
> | FAENet (w/o pooling)     | 0.486±0.010   | 0.834±0.006     | 0.703±0.004    | OOM            |
> | DimeNet (w/o pooling)    | 0.467±0.010   | 0.797±0.012     | 0.712±0.004    | OOM            |
> | GVP (w/o pooling)        | 0.467±0.010   | 0.797±0.012     | 0.744±0.004    | OOM            |
> | FastEGNN (w/o pooling)   | 0.477±0.005   | 0.816±0.014     | 0.753±0.001    | OOM            |
> | **3D model (with nuc. pooling)** |         |                 |                |                |
> | EGNN (nuc. pooling)      | 0.364±0.003   | 0.619±0.007     | 0.663±0.010    | OOM            |
> | SchNet (nuc. pooling)    | 0.390±0.006   | 0.685±0.006     | 0.655±0.038    | OOM            |
> | FastEGNN (nuc. pooling)  | 0.444±0.003   | 0.753±0.015     | 0.710±0.011    | OOM            |
>
>
> &nbsp;
>
> Please let us know if there are any further questions. Thanks a lot!

---

> ### Comment · Reviewer_HBUE · 2024-11-22
>
> Thank you very much for the authors' replies. I still have some questions and hope to get their explanations.
> > **D1. Need to add original improvements to the model.**
>
> The facts stated in the authors' response are indeed the core issues in this field. High-quality RNA datasets are very rare, so the benchmark compilation contribution of this article is indeed very important.
>
> However, I and several other reviewers are very concerned about the fact that ICLR, as a conference in the field of machine learning, may not contribute enough to just propose benchmarks and test existing baselines (although this article belongs to the field of "datasets and benchmarks"). Although this article compares a large number of baselines, it lacks the authors' own innovation in model architecture. Generally, AI for Science benchmarks are more or less designed for special scenarios (e.g. GVP-GNN and GNS). Can the authors take advantage of the particularity of RNA to add some more. For example, introduce some small sample learning techniques (such as active learning, etc.) based on the lack of data? Or can they improve FastEGNN based on the sequence structure of RNA?
>
> > **D2. About FastEGNN settings.**
>
> Can the authors list the specific parameters of FastEGNN? For example, how many virtual nodes are used? Is there any additional edge deletion?
>
> As far as I know, it seems that the virtual nodes of FastEGNN are introduced without prior knowledge, and the point set distribution of the experimental data set in the original paper does not show obvious serialization characteristics like RNA. Can the poor performance of FastEGNN on RNA data be considered to be due to the contribution of virtual nodes to irrelevant real nodes? Can the authors further analyze such phenomena?
>
> In response to this problem, can the authors modify the initialization of virtual nodes (for example, set one for each RNA substructure) and the way virtual nodes are connected to real nodes (delete edges with a long distance or introduce radial basis functions) as their customized model on the benchmark? I think that designing a methodology that uses RNA prior knowledge to modify the general model into a customized model can be a good contribution to match the benchmark to make the paper more acceptable.
>
> > **D3. More baselines.**
>
> The method of using high-degree steerable features is also very important for 3D geometric graph networks. As a benchmark, I think it would be more beneficial to the quality of the article to choose one from TFN, NequIP, MACE, EquiformerV2 for testing. I recommend using TFN and using the code in https://github.com/chaitjo/geometric-gnn-dojo/blob/main/models/tfn.py, which is easier to modify.
> In addition, it may not be particularly easy to migrate special models such as MEAN and dyMEAN. I can also understand that the authors do not have enough time to design them specifically, so I will not ask for them.

---

> > ### Author Response · Authors · 2024-11-22
> > **Author Response**
> >
> > Thanks for your comments. Below are our responses:
> >
> > **RD1. Original improvements to the baseline models.**
> > We thank the reviewer for acknowledging the contribution of our proposed datasets and the utmost importance of benchmarking for RNA property prediction. However, we must disagree with the reviewer's claims that "datasets and benchmarks" papers require contribution in improving the existing baselines. We kindly refer the reviewer to the ICLR "call for papers" (https://iclr.cc/Conferences/2025/CallForPapers) where "datasets and benchmarks" unambiguously stand as a separate category. To emphasize this further, please also see highly impactful and well-cited recent papers in the biology domain which were accepted at top conferences (including ICLR) under the "datasets and benchmarks" subject area [1, 2, 3] and focused purely on new datasets and solid benchmarks, an essential and first and foremost requirement for novel methodology development and evaluation for future research.
> >
> >
> >
> > **RD2. About FastEGNN settings.**
> > Following Table 1 of the original FastEGNN paper [4], we fixed the number of virtual nodes to 3. Edges were not removed; instead, we used a threshold to infer edge existence across all baselines. This threshold was treated as a tunable hyperparameter.
> >
> > We agree that improving FastEGNN for RNA is indeed a promising direction for future work. However, we would like to point out that this is out of the scope of this work for the following reasons.
> >
> > 1) This paper aims to establish high-quality datasets and benchmarks, which as the reviewer notices, are very rare for the RNA domain. We believe that establishing the benchmark and providing experimental baselines is the critical prerequisite before developing new models or modifying existing ones, and thus is a strong contribution in itself.
> >
> > 2) The original FastEGNN paper limits the use of virtual nodes to a maximum of 10, with these virtual nodes fully connected to each other. Assigning a virtual node to each nucleotide would introduce $n^2$ additional edges for an RNA sequence with $n$ nucleotides, significantly increasing the model’s computational requirements. Significantly modifying FastEGNN to overcome these limitations in itself is a non-trivial task and will warrant a separate paper.
> >
> >
> > **RD3. More baselines.**
> > High-degree steerable methods are indeed an important category within 3D equivariant models. However, they typically rely on spherical harmonics which are computationally complex and resource-intensive. While we considered these steerable methods, they ran out of memory due to the length of RNA sequences. Specifically, we used the TFN code provided in the link you mentioned, but **it encountered out-of-memory (OOM)**. Similarly, MACE also faced OOM issues. This is also an important insight enabled by our paper that spherical models, such as TFN and MACE, which show promise in small molecule modeling domains are not out-of-the-box applicable for larger molecules such as RNA. This is an insight that will benefit the practitioners for real-world applications and we are happy to include it in our paper, if you suggest.
> >
> > Also, we would like to point out that in prior work [5] (Figure 1 and is also mentioned in the Github link you provided), GVP-GNN is the most complex model which still allows modeling larger molecules such as RNA sequences and we have included it already in our revised version.
> >
> > &nbsp;
> >
> > Please let us know if you have other questions. We hope the reviewer will consider a more positive evaluation of our work.
> >
> > &nbsp;
> >
> > References.
> > [1] FLIP: Benchmark tasks in fitness landscape inference for proteins, NeurIPS 2021.
> > [2] BEND: Benchmarking DNA Language Models on Biologically Meaningful Tasks, ICLR 2024.
> > [3] GeneDisco: A Benchmark for Experimental Design in Drug Discovery, ICLR 2022.
> > [4] Improving Equivariant Graph Neural Networks on Large Geometric Graphs via Virtual Nodes Learning
> > [5] On the Expressive Power of Geometric Graph Neural Networks.

---

> ### Comment · Reviewer_HBUE · 2024-11-24
>
> I have carefully read the author's feedback and think that the author's explanation makes sense. I have combined the opinions of both parties and put forward the following suggestions for improvement. If the author can do this, I will consider recommending that this article be accepted.
>
> > **D4. About models using high-degree steerable features**
>
> I still think it is necessary to use these models based on high-degree steerable features as baselines. After all, they are also an important part of equivariant models. Authors can avoid OOM problems by adjusting batch_size or reducing channels. Alternatively, authors can consider using the recently emerged SO3krates [a] or HEGNN [b] to introduce high-degree steerable features while avoiding the complexity of tensor products. I understand that the remaining time in the current discussion stage may not be enough to complete the experiment, so in the discussion stage, I only require the authors briefly **discuss the possible benefits of high-degree steerable features** in the article. Presumably, there is still enough time from acceptance to submission of camera ready for the authors to supplement the experimental results (and also the results of models like SaVeNet, LEFTNet and MEAN).
>
> [a] Frank J T, Unke O T, Müller K R, et al. A Euclidean transformer for fast and stable machine learned force fields[J]. Nature Communications, 2024, 15(1): 6539.
>
> [b] Cen J, Li A, Lin N, et al. Are High-Degree Representations Really Unnecessary in Equivariant Graph Neural Networks?[J]. arXiv preprint arXiv:2410.11443, 2024.
>
> > **D5. Discussion on the inapplicability of FastEGNN**
>
> I think the authors' response makes a lot of sense, and I hope to include this **discussion** in the manuscript. I think FastEGNN is an excellent model, although its CoM virtual node initialization may not be suitable for this problem that clearly has a sequence prior. It would be great if the authors could combine this to explain why FastEGNN performs poorly and give suggestions for improvement. I believe this will not only improve the value of this article, but also be a respect for the original author of the FastEGNN model.
>
> > **D6. Discussion on considering noise in model design**
>
> Since most studies in this field do not explicitly address noise when designing model architectures, could the authors **discuss how to effectively account for noise during model design** and what methods might be employed to minimize its impact?

---

> > ### Author Response · Authors · 2024-11-25
> >
> > **RD4. About models using high-degree steerable features**
> >
> > Thanks for your suggestions. We actually tried to use a smaller batch size. In the paper, we used batch size=96 consistently for all baselines. But for methods using spherical harmonics, even batch size = 16 or 8 faces OOM issues.
> >
> > Thank you for the nice suggestion about HEGNN and SO3krates. We have now added a section Appendix E.1 of the paper and referred to  main text (lines 537-538) to discuss the possible benefits of high-degree steerable features. And we will continue to run more baselines mentioned by the reviewer to make the benchmark more solid and update it in the camera-ready version.
> >
> > **RD5. Discussion on the inapplicability of FastEGNN**
> >
> > Thanks for your understanding. We added the discussion about FastEGNN in our paper (see lines 295 - 299). We appreciate FastEGNN’s significant contributions to this field and believe it will be a future direction to explore.
> >
> >
> > **RD6. Discussion on considering noise in model design**
> >
> > Based on the analysis in our paper, one of the most effective ways to deal with noise data may be the ensemble method. Our experiments show that though the 1D method does not perform well on clean datasets, the 1D method is more robust to noise. An effective way to handle noise can be through ensemble methods. For instance, combining 1D, 2D, and 3D models by independently learning representations and integrating them via attention mechanisms that dynamically weigh each modality based on task relevance and noise can leverage the robustness of 1D methods while benefiting from the strengths of 2D and 3D approaches and be a good direction for future research.
> >
> > We included this part in the Appendix E.2 in our paper and referred to it main text (lines 535-537).
> >
> > &nbsp;
> >
> > We hope this addresses your concerns and helps enhance our paper.

---

> ### Comment · Reviewer_HBUE · 2024-11-25
>
> Dear authors, thank you for your responses. Apart from the misplaced references of SO3krates and HEGNN in Line. 1088, I think your summary is quite accurate. I raise my rating to "Weakly Accepted".

---

> > ### Author Response · Authors · 2024-11-26
> > **Thank your for updating your score**
> >
> > Thank you very much for your useful feedback, kind understanding and raising your score!

---

> > ### Author Response · Authors · 2024-12-03
> >
> > Dear Reviewer,
> >
> > With several hours remaining until the rebuttal deadline, we hope we have successfully addressed your concerns and questions. If you find our responses satisfactory, we kindly ask you to consider raising your score.
> >
> > We truly appreciate your time, effort, and valuable contributions to improving our work.
> >
> > Best regards,
> > The Authors.

---

> ### Comment · Reviewer_HBUE · 2024-12-03
>
> I carefully compared this article with other benchmark articles according to the requirements of the benchmark and reviewed it again. In fact, the amount of work in this article is sufficient, and as far as I know, it is the first to study the properties of RNA in a real-world scenario. In particular, large-scale long-chain RNA datasets are rare, and the author's integrated representation modalities and combed research directions are valuable. By reading 'Author Responses to Reviewer JSSw', I was further convinced of the value of this article, so I upgraded the rating to 'Clear Accepted'.

---

> ### Author Response · Authors · 2024-12-03
> **Thank you for strongly advocating for our work!**
>
> We thank the reviewer for raising their score and championing for our work highlighting our contributions as novel and meriting publication owing to the utility of the real-world datasets and experiments for the field!

---

### Official Review · Reviewer_k6pU · 2024-10-30

**Soundness:** 2
**Presentation:** 3
**Contribution:** 2
**Rating:** 5
**Confidence:** 4

**Summary:**

The paper provides a curated set of RNA datasets with annotated 2D and 3D structures and investigates RNA property prediction using various deep learning models, focusing on 1D (nucleotide sequences), 2D (graph representations), and 3D (atomic structures) approaches. Key findings reveal that 2D models generally outperform 1D models, while 3D models excel in noise-free scenarios but are sensitive to noise. In contrast, 1D models show greater robustness in noisy and OOD conditions. The authors emphasize the trade-offs of each approach and advocate for future research to integrate the strengths of all models.

**Strengths:**

1. This study provides a thorough comparison of 1D, 2D, and 3D models, showcasing their respective strengths and weaknesses in handling RNA data.
2. This study provides a comprehensive analysis of various deep learning models, assessing their performance under different conditions, including limited data and labels, different types of sequencing errors, and out-of-distribution scenarios, which is crucial for real-world applications.
3. The commitment to transparency and reproducibility by making methodologies, datasets, and code publicly available promotes collaborative progress in the field.

**Weaknesses:**

1. The article lacks methodological innovation, missing deep improvements on existing technologies and novel algorithm designs.
2. The article merely compares various metrics, and the key points are not sufficiently emphasized.
3. Both the secondary and tertiary structures are predicted using software, particularly the tertiary structure, which is not very accurate. This can lead to significant uncertainties in further property predictions.

**Questions:**

1. In recent years, many RNA language models have been proposed, applicable to various downstream tasks. Compared to these models, what are the advantages and disadvantages of the methods mentioned in the article?
2. What are the criteria for selecting secondary structure prediction tools, and is there a detailed analysis and further experimentation?
3. Please explain the motivation behind the five task settings, as these experiments may make the article appear cumbersome and the key points unclear.

---

> ### Author Response · Authors · 2024-11-21
> **Author Responses (1/2)**
>
> We thank the reviewer for their feedback and appreciating the thoroughness and comprehensiveness of our experiments and for highlighting the value of our newly annotated RNA data. We address the questions and comments raised by the reviewer point-by-point:
>
> **RW1: Lacking methodological innovation:**
> We appreciate the reviewer's observation about the simplicity of the proposed framework. We would like to emphasize the our paper sits under “datasets and benchmarks”, a subject area highlighted in ICLR call for papers. Our main contributions are: 1) introducing first of its kind RNA datasets with all 1D, 2D and 3D structures and property labels; 2) providing a modular unified testing environment for benchmarking 1D, 2D and 3D property prediction models; 3) studying existing 1D, 2D and 3D models to assess the impact of geometric information in a range of real-world scenarios and establish baselines for future research in this direction. No such study exists to date despite the importance of RNA as therapeutic modality.
>
>
> **RW2: The article merely compares various metrics, and the key points are not sufficiently emphasized.**
> In this work, we provide a comprehensive comparison beyond just RNA property prediction accuracy, which includes a range of real-world challenges of modeling RNA: noise robustness, OOD generalization, data and label efficiency. We derived several key insights including: 2D spectral methods outperforming 1D and 3D methods in low-to-moderate noise regimes (highlighted in bold in line 259 and line 373); 3D models outperforming 1D models under limited data regime even despite the structural noise (highlighted in bold in line 383); 1D sequence methods require 2x-5x more training data to match the performance of 2D and 3D methods (line 379), but excel in high noise regime (highlighted in bold in line 462); simple Transformer1D2D with soft structural prior outperforms the rest of the models even in high noise regimes while still utilizing structural information (lines 472-476). We consider these insights significant since they cover a range of real-world scenarios that have not been investigated in prior work. We would appreciate the suggestions from the reviewer on which key points require further elaboration and more emphasis.
>
> **RW3: Impact of uncertainties in secondary and tertiary structures for property predictions:**
> We wish to clarify that this is intentional and corresponds to the real-world setting reflecting the challenges practitioners face when modeling RNA properties. We are not aware of any dataset containing triplets of 1D RNA sequences, experimentally determined 2D and 3D structures, and corresponding experimentally measured property labels. This is due to the high cost and technical difficulty of experimentally determining RNA structures and measuring their properties. Consequently, practitioners must rely on predicted structures or fall back to 1D sequence models. Thus, the structural uncertainties are inherent and inevitable in practice and including them into analysis better reflects practical reality.

---

> ### Author Response · Authors · 2024-11-21
> **Author Responses (2/2)**
>
> **RQ1: Results with RNA language models:**
> Good suggestion! We have now **benchmarked 2 SOTA RNA language models**: **RNA-FM** [1] and **SpliceBERT** [2] and added the results in Table 1 in the main text (highlighted in blue). We find that for 2 out of 4 datasets (Covid and Ribonanza-2k), RNA foundation models perform worse than the supervised transformer baseline wheres for the other two datasets (Tc-Riboswitches and Fungal), they achieve similar performance (results reported n MCRMSE, lower is better). This is consistent with recent papers in multiple biology domains demonstrating generalized foundation models are yet to surpass specialized supervised baselines (see [3,4,5]). Thus, all our presented conclusions hold.
>
> | Model                     | COVID         | Ribonanza       | Tc-Ribo        | Fungal         |
> |---------------------------|---------------|-----------------|----------------|----------------|
> | **1D model**             |               |                 |                |                |
> | Transformer1D            | ***0.361±0.017***   | ***0.705±0.015***     | 0.705±0.019    | ***1.417±0.005***    |
> | RNA-FM                   | 0.591±0.081   | 0.909±0.144     | ***0.693±0.001***    | 1.420±0.028    |
> | SpliceBERT               | 0.588±0.077   | 1.022±0.144     | 0.708±0.003    | 1.435±0.059    |
>
>
> &nbsp;
>
>
> **RQ2: Criteria for selecting secondary structure prediction tools:**.
> We choose EternaFold as the tool for secondary structure prediction because of the reported SOTA performance of EternaFold in recent works ([6,7,8]) and has been explained in lines 143-146 of the main text.
>
> **RQ3: Explain the motivation behind the five task settings:**
> We appreciate the reviewer's feedback and recognize the importance of clearly explaining the motivation for the selected tasks. These tasks were specifically chosen to simulate real-world challenges encountered in RNA property prediction, focusing on diverse practically relevant aspects of model evaluation. The tasks aim to address critical questions such as:
> **Task 1:** how effectively models leverage structural information of RNA which so far has not been explored in literature in the context of RNA property prediction;
> **Tasks 2 and 3:** how well they perform with limited training data or partial labels, a common constraint in experimental settings due to the costs of obtaining large experimental databases;
> **Tasks 4 and 5:** the robustness of the models under sequencing noise, which mirrors variations and errors produced by RNA sequencing platforms and methods.
> In Sec 3 (pages 4 and 5), we have cited relevant works which highlight exactly these scenarios for RNA data. We have now also added **detailed descriptions and motivations for these tasks in the Appendix D (pages 19-20 and highlighted in blue)** and referred to it in main text (lines 248-249).
>
> &nbsp;
>
> Please let us know if there are any further questions. Thanks a lot!
>
> &nbsp;
>
> References:
> [1] Interpretable RNA Foundation Model from Unannotated Data for Highly Accurate RNA Structure and Function Predictions, Chen et al., arXiv 2022.
> [2] Self-supervised learning on millions of primary RNA sequences from 72 vertebrates improves sequence-based RNA splicing prediction, Chen et al., Briefings in Bioinformatics, 2023.
> [3] Specialized Foundation Models Struggle to Beat Supervised Baselines, Xu, Gupta et al., FM4Science@NeurIPS 2024
> [4] Convolutions are competitive with transformers for protein sequence pretraining, Yang et al., cell Systems, 2024
> [5] Assessing the limits of zero-shot foundation models in single-cell biology, Kedzierska et al., bioRxiv, 2023
> [6] RNA secondary structure packages evaluated and improved by high-throughput experiments, Wayment-Steele et al., Nature Methods, 2022.
> [7] Deep learning models for predicting RNA degradation via dual crowdsourcing, Wayment-Steele et al., Nature Machine Intelligence, 2022
> [8] Ribonanza: deep learning of RNA structure through dual crowdsourcing, He et al., bioRxiv, 2024

---

> ### Author Response · Authors · 2024-11-25
> **Friendly request for feedback on responses to your questions...**
>
> Dear Reviewer,
>
> Thank you for your valuable feedback. We have carefully responded to all your concerns and would be grateful if you could consider raising your score.
>
> As the rebuttal deadline approaches, we would greatly appreciate your feedback at your earliest convenience. Thank you again for your time and thoughtful suggestions!
>
> Best regards,
> The Authors

---

> > ### Author Response · Authors · 2024-12-01
> >
> > As we approach the end of the discussion period, we kindly request your review of our detailed responses and revisions, which aims at comprehensively addressing all the concerns you raised. We respectfully request if would you be willing to increase your score if you are happy with how we have addressed your suggestions? Thank you!

---

> > > ### Author Response · Authors · 2024-12-03
> > >
> > > Dear Reviewer,
> > >
> > > With several hours remaining until the rebuttal deadline, we hope we have successfully addressed your concerns and questions. If you find our responses satisfactory, we kindly ask you to consider raising your score.
> > >
> > > We truly appreciate your time, effort, and valuable contributions to improving our work.
> > >
> > > Best regards,
> > > The Authors.

---

### Official Review · Reviewer_JSSw · 2024-11-03

**Soundness:** 2
**Presentation:** 3
**Contribution:** 2
**Rating:** 6
**Confidence:** 4

**Summary:**

This paper introduce a newly curated set of RNA datasets with enhanced 2D and 3D structural annotations, providing a resource for model evaluation on RNA data. The paper reveals that models with explicit geometry encoding generally outperform sequence-based models, and geometry-unaware sequence-based models are more robust under sequencing noise but often require around 2 − 5× training data to match the performance of geometry-aware models. The authors conducted thorough and detailed experiments to support their proposed arguments.

**Strengths:**

1. The authors investigated the enhancement of RNA property prediction through the utilization of both 2D and 3D data, and explored the performance degradation of corresponding models under various influencing factors, including noisy data and partial label.

2. The authors collected a substantial amount of RNA sequence data across a wide range of nucleotide count intervals, ensuring comprehensive coverage.

3. The authors conducted extensive experiments using various models on the dataset to support their proposed arguments.

Originality, quality, clarity, and significance:

The paper is original and provides a comprehensive exploration of the impact of different types of input data on RNA property prediction capabilities for the first time. The writing is clear, and the overall quality is marginally acceptable. This research makes a contribution to the exploration of RNA property prediction.

**Weaknesses:**

1. The authors utilized a limited number of 3D models for geometric structure modeling, most of which are relatively early models, and neither of the two models (EGNN and SchNet) is specifically designed for 3D RNA structure modeling. Therefore, I believe their performance does not fully reflect the potential improvements offered by geometric information across various datasets. The authors are supposed to validate models specifically designed for RNA 3D structure modeling, such as ARES [1] and PaxNet [2], as well as some classic models in the protein domain (like GVP [3], GearNet [4], and MEAN [5]).

2. The pooling strategy is difficult to classify as an innovative contribution from the authors, as it is relatively simple. Thus, the models employed in the paper are essentially existing models, and the authors have not proposed their own model. Since 3D data encompasses more information than 1D and 2D data, the authors should reflect on how to better utilize 3D information to enhance model performance.

3. As noted in the paper, "all-atom SchNET and EGNN rely on a limited local neighborhood of adjacent atoms, limiting their receptive fields and preventing them from capturing long-range dependencies", full-atom modeling indeed incurs a significant computational burden. The authors could consider adopting the approach from methods like MEAN [5], treating each nucleotide as a node in a graph, which would allow for the expansion of local neighborhood relationships. Alternatively, the strategy used in PaxNet [2] could be employed to model the long-range and short-range interactions.

4. The analysis in section 4.2 lacks insights. Some conclusions are quite obvious, such as the enhancement of performance due to increased training data, which is even more pronounced in transformer-like structures. Moreover, the inferior performance of 3D models compared to 2D models can be attributed to the significantly lower prediction accuracy of existing methods for 3D structures compared to 2D structures.

5. The analysis in section 4.3 presents similar issues. Since the authors introduced sequencing noise, this error accumulates greater noise in both 2D and 3D data as a result of using 2D and 3D predictive tools. Therefore, it is expected that the transformer1D, which directly models 1D sequence data, would exhibit stronger performance.

[1] Geometric deep learning of RNA structure, Science 2021.

[2] Physics-aware Graph Neural Network for Accurate RNA 3D Structure Prediction, NIPS 2022 workshop.

[3] Learning from Protein Structure with Geometric Vector Perceptrons, ICLR 2021.

[4] Protein representation learning by geometric structure pretraining, ICLR 2023.

[5] Conditional Antibody Design as 3D Equivariant Graph Translation, ICLR 2023.

**Questions:**

1. Predicting the 3D structure of models directly from 1D sequence data can indeed introduce noise, affecting model performance. Have the authors considered finding a 3D RNA dataset, such as the RNAsolo [1]?

2. This paper primarily focuses on predicting RNA properties and utilizes the MCRMSE metric. I would like to know what specific properties are being referred to, and what are the units for the predicted values?

[1] Rnasolo: a repository of cleaned pdb-derived rna 3d structures, Bioinformatics 2022.

**Details Of Ethics Concerns:**

The copyright of the collected datasets.

---

> ### Author Response · Authors · 2024-11-21
> **Author Responses (1/2)**
>
> We thank the reviewer for their feedback and appreciating the originality, thoroughness, and comprehensiveness of our experiments that highlight the value of our newly annotated RNA datasets. We also appreciate the reviewer's observation that our work presents a first study of its kind for RNA property prediction in range of real-world scenarios.
>
> Now we address each weakness and question raised by the reviewer.
>
> **RW1. Experiments with more 3D models:**
> Following the reviewer's suggestion, we have now added a comparison of four additional 3D models (GVP, DimeNet, and recent FAENet and FastEGNN) in Table 1 and Figs. 2, 3, 5, 6 in the main text (highlighted in blue). However, we still find that all these models show similar performance on RNA property prediction tasks with 3D models failing to outperform 2D models. We further choose FastEGNN for all subsequent experiments and observe the similar trend as reported earlier with EGNN and SchNet.
>
>
> | Model                     | COVID         | Ribonanza       | Tc-Ribo        | Fungal         |
> |---------------------------|---------------|-----------------|----------------|----------------|
> | **1D model**             |               |                 |                |                |
> | Transformer1D            | 0.361±0.017   | 0.705±0.015     | 0.705±0.019    | 1.417±0.005    |
> | **2D model**             |               |                 |                |                |
> | Transformer1D2D          | 0.305±0.012   | 0.514±0.004     | 0.633±0.001    | OOM            |
> | GCN                      | 0.359±0.009   | 0.509±0.004     | 0.640±0.005    | 1.192±0.077    |
> | GAT                      | 0.315±0.006   | 0.534±0.006     | 0.603±0.004    | 1.112±0.035    |
> | ChebNet                  | 0.279±0.015   | 0.499±0.005     | 0.599±0.001    | 1.018±0.023    |
> | Graph Transformer        | 0.318±0.008   | 0.500±0.005     | 0.604±0.001    | 1.317±0.002    |
> | GraphGPS                 | 0.332±0.013   | 0.523±0.003     | 0.610±0.012    | 1.025±0.081    |
> | **3D model (w/o pooling)** |               |                 |                |                |
> | EGNN (w/o pooling)       | 0.480±0.025   | 0.808±0.023     | 0.725±0.002    | OOM            |
> | SchNet (w/o pooling)     | 0.499±0.030   | 0.843±0.004     | 0.704±0.001    | OOM            |
> | FAENet (w/o pooling)     | 0.486±0.010   | 0.834±0.006     | 0.703±0.004    | OOM            |
> | DimeNet (w/o pooling)    | 0.467±0.010   | 0.797±0.012     | 0.712±0.004    | OOM            |
> | GVP (w/o pooling)        | 0.467±0.010   | 0.797±0.012     | 0.744±0.004    | OOM            |
> | FastEGNN (w/o pooling)   | 0.477±0.005   | 0.816±0.014     | 0.753±0.001    | OOM            |
> | **3D model (with nuc. pooling)** |         |                 |                |                |
> | EGNN (nuc. pooling)      | 0.364±0.003   | 0.619±0.007     | 0.663±0.010    | OOM            |
> | SchNet (nuc. pooling)    | 0.390±0.006   | 0.685±0.006     | 0.655±0.038    | OOM            |
> | FastEGNN (nuc. pooling)  | 0.444±0.003   | 0.753±0.015     | 0.710±0.011    | OOM            |
>
>
>
> Regarding 3D models such as ARES and PaxNet, we note that they are meant for RNA 3D structure prediction and ranking and do not support RNA property prediction out of the box. Similarly, methods such as MEAN are highly specialized for modeling antibody domains by design and adapting them for RNA property prediction will be a non-trivial contribution in itself.
>
> **RW2 \& RW3: Pooling strategy and dealing with long-range dependencies:**
> We appreciate the reviewer's observation about the simplicity of the proposed framework. We would like to emphasize the our paper sits under **“datasets and benchmarks”**, a subject area highlighted in ICLR call for papers. Our main contributions are: 1) introducing first of its kind RNA datasets with all 1D, 2D and 3D structures and property labels; 2) providing a modular unified testing environment for benchmarking 1D, 2D and 3D property prediction models; 3) studying existing 1D, 2D and 3D models to assess the impact of geometric information in a range of real-world scenarios and establish baselines for future research in this direction. No such study exists to date despite the importance of RNA as therapeutic modality.
>
> With this, substantially modifying existing or developing novel 3D methods is outside the scope of this work. At the same time, we agree with the reviewer that better utilization of 3D information is important, and our work will serve to foster future research in this direction.

---

> ### Author Response · Authors · 2024-11-21
> **Author Responses (2/2)**
>
> **RW4: Conclusions from data efficiency experiment in Section 4.2:**
> The experiments in section 4.2, investigating models' data efficiency, not only reveal that the performance improves with increasing training data, but also uncovers novel insights of 2D spectral models excelling in low data and partial label regimes, and 3D models outperforming 1D models under limited data regime despite the structural noise. To the best of our knowledge, these insights are novel. We consider these insights significant in the real-world scenarios where large fully-labeled datasets are unavailable.
>
> **RW4: lower prediction accuracy of existing methods for 3D structures:**
> To the best of our knowledge, no datasets exist containing triplets of 1D RNA sequences, experimentally determined 2D/3D structures, and corresponding property labels, as discussed in Appendix A. This is due to the high cost and technical difficulty of experimentally determining RNA structures and measuring their properties. Consequently, practitioners must rely on predicted structures or fall back to 1D sequence models. Thus, the structural inaccuracies are often inherent and inevitable in practice. At the same time, as the reviewer rightly noted, 3D data contains more information than 1D and 2D data. However, before our work, it was unclear whether this extra, potentially noisy information would add value for property prediction compared to 1D or 2D data. No other prior work has investigated this trade-off between richer information content and potentially more uncertain 3D structures.
>
>
> **RW5: Conclusions from sequencing noise experiment in Section 4.3:**.
> We want to highlight that the goal of experiments in Section 4.3 was to assess the models under \textit{realistic} sequencing noise. To this end, we sampled the noise profiles practically observed in real-world sequencing platforms (line 430-434). Since no prior work has studied the impact of realistic sequencing noise on quality of predicted structures, it was not clear how different classes of property prediction models perform in this realistic scenario and to what degree their performance deteriorates under realistic noise. Our experiment not only reveals that Transformer1D is more robust under high sequencing noise, but also that 2D models still perform the best under low-to-moderate noise regimes (now highligted in blue in lines 479-480). Additionally, we reveal that simple Transformer1D2D with soft structural prior outperforms the rest of the models even in high noise regimes while still utilizing structural information (lines 473-476). We consider these insights significant since they cover real-world deployment scenarios and no prior work has investigated the impact of \textit{realistic} sequencing noise on 1D, 2D and 3D property prediction methods.
>
>
> **RQ1. Have the authors considered finding a 3D RNA dataset, such as the RNAsolo?**
> We thank the reviewer for the suggestion. However, note that RNAsolo dataset does not include any property labels associated with these structures and hence cannot be used for our purpose of studying the impact of sequence and structural information for property prediction tasks. Currently, we are not aware of any dataset containing triplets of 1D RNA sequences, experimentally determined 2D and 3D structures, and corresponding experimentally measured property labels per data point. This also highlights the importance of our work in a real-world setting where both experimentally determined geometric structures and property labels are unavailable.
>
>
> **RQ2. What specific properties are being referred to, and what are the units for the predicted values?**
> We have described the properties in Subsection 2.1 of the main text. To summarize, the properties we model are Riboswitch switching behavior for the Tc-Riboswitches dataset, nucleotide degradation for the COVID-19 dataset, reactivity for the Ribonanza-2k dataset, and expression for the Fungal dataset.
>
> For  Tc-Riboswitches, the labels are percentage reflecting switching behaviors. For COVID-19, Ribonanza-2k datasets, the labels degradation and reactivity labels are normalized intensity (hence without units) and for Fungal dataset, the expression labels are measured in transcripts per million (TPM) per kilobase million (RPKM).
>
> &nbsp;
>
> Please let us know if there are any further questions. Thanks a lot!

---

> ### Author Response · Authors · 2024-11-21
> **Request for Clarification on Score-Comment Alignment**
>
> Dear Reviewer JSSw,
>
> We noticed that your review mentions, “the overall quality is marginally acceptable,” yet the final score assigned is “reject.” We kindly ask if you could check the score to ensure alignment with your comments, and consider our rebuttal at the same time.
>
> Thank you for your time and consideration.
>
> Best regards,
> The Authors

---

> ### Author Response · Authors · 2024-11-25
> **Friendly request for feedback on responses to your questions...**
>
> Dear Reviewer,
>
> Thank you for your valuable feedback. We have carefully responded to all your concerns. Could you kindly review your rating and consider our rebuttal? We would be grateful if you could consider raising your score.
>
> As the rebuttal deadline approaches, we would greatly appreciate your feedback at your earliest convenience. Thank you again for your time and thoughtful suggestions!
>
> Best regards,
> The Authors

---

> > ### Comment · Reviewer_JSSw · 2024-11-30
> >
> > Thank you for your response. Some of my concerns have been addressed, so I decide to raise my score.

---

> > > ### Author Response · Authors · 2024-12-03
> > >
> > > Dear Reviewer,
> > >
> > > With several hours remaining until the rebuttal deadline, we hope we have successfully addressed your concerns and questions. If you find our responses satisfactory, we kindly ask you to consider raising your score.
> > >
> > > We truly appreciate your time, effort, and valuable contributions to improving our work.
> > >
> > > Best regards,
> > > The Authors.

---

> ### Author Response · Authors · 2024-11-30
>
> We thank the reviewer for raising their score based on our response. If the reviewer can point us to which concerns are still remaining so that we can address them from our side to help raise their score even further. Thank you!

---

> ### Author Response · Authors · 2024-12-03
> **About Ethics Concerns and Request To Reconsider Score**
>
> In this paper, we collect RNA sequences from publicly available datasets and generate the 2D and 3D structures from publicly available tools. We cite all the sources in our paper, please refer to Section 2.1 and 2.2 in our paper. Therefore, we don't see any potential ethical issues of the data used in this paper.
>
> We have also mentioned this in the "Ethics Statement" part in our paper. If all your concerns are addressed, could you please consider raising your score as only few hours of discussion period remain?

---

> > ### Comment · Reviewer_JSSw · 2024-12-03
> >
> > Thank you for addressing the ethics concerns. Most of my concerns have been addressed, and I raise the score to 6.

---

> > > ### Author Response · Authors · 2024-12-04
> > > **Thank you for raising your score!**
> > >
> > > We thank the reviewer for raising their score and acknowledging our rebuttal responses! We appreciate your valuable suggestions during the discussion process which helped improve our work.

---

### Author Response · Authors · 2024-12-04
**General summary of revisions and rebuttal responses**

Dear Reviewers and Area Chair,

We sincerely thank all reviewers for their valuable feedback and thoughtful comments on our submission. We are pleased to see recognition of the key strengths of our work and would like to summarize the major advantages of our work highlighted by the reviewers and discussed in our rebuttal responses:
1. ***Introduction of New Datasets:*** A key contribution of our work is the creation and curation of first-of-its-kind diverse datasets of RNA sequences (1D) annotated with 2D, and 3D structures for various property prediction tasks. These datasets cover multiple species, sequence lengths, and application areas, and provide an essential foundation for evaluating future models in an underexplored field of RNA modeling with immense therapeutic potential. No prior work offers datasets with such comprehensive structural annotations (1D sequence, 2D secondary structure, 3D all-atom) and property labels.

2. ***Extensive Evaluation of Structural Information for RNA Property Prediction:***  We establish a unified modular testing environment encompassing 15 representative and state-of-the-art models for 1D, 2D, and 3D RNA property prediction. During the rebuttal, we expanded our analysis by including additional state-of-the-art 3D baselines and 1D RNA language models, further validating our conclusions and providing a useful resource for the community to benchmark and develop future RNA modeling approaches. This framework not only facilitates direct comparison of diverse methods but also highlights key performance trade-offs across different structural representations and modeling scenarios.

3. ***Addressing Important Real-World RNA Modeling Challenges:***  Beyond prediction accuracy, we tackled real-world challenges practitioners routinely face such as noise robustness, OOD generalization, data/label efficiency, computational efficiency, and model scalability. Key insights include: (1) 2D spectral methods outperform 1D and 3D in low-to-moderate noise, (2) 3D models excel with limited data, even under structural noise when scaled up using RNA specific biological priors, (3) 1D methods require 2x-5x more data to match 2D/3D but perform best in high noise while 2D models outperforming other models for low noise levels, and (4) adding soft structural priors to 1D models (e.g.-Transformer1D2D) outperforming more complex models across noise regimes. These findings address previously unexplored scenarios and reflect practical challenges in working with diverse RNA sequences, offering actionable insights into the interplay of geometric features and advancing the field.

We appreciate the reviewers’ engagement and constructive feedback, which allowed us to refine our work further. We believe these strengths, combined with the revisions and additional experiments provided, demonstrate the significance and impact of our contributions.

Thank you for your time and consideration.

Best regards,

The Authors

---

### Meta-Review · Area_Chair_pXt6 · 2024-12-18

**Metareview:**

This paper introduces a curated set of RNA datasets with enhanced 2D and 3D structural annotations, providing a valuable resource for evaluating RNA property prediction models. It systematically investigates the impact of incorporating geometric information, showing that models with explicit geometry encoding generally outperform sequence-based models, while the latter are more robust under noise but require more training data. The paper emphasizes the trade-offs between 1D, 2D, and 3D models, with 2D models typically outperforming 1D ones, and 3D models excelling in noise-free conditions but being sensitive to noise.

Strengths:

1. The study compiles a large and diverse set of RNA sequence data across a broad range of nucleotide count intervals, ensuring comprehensive coverage.

2. This work provides an in-depth comparison of 1D, 2D, and 3D models, conducting a thorough analysis of various deep learning approaches and evaluating their performance under different conditions, which is essential for real-world applications.

3. The authors demonstrate a strong commitment to transparency and reproducibility by making their methodologies, datasets, and code publicly available, fostering collaborative progress in the field.

Weaknesses:

1. The methods employed are relatively outdated, with recent advances in 1D sequence models and 2D topological graph representations not fully addressed. Additionally, newer works on 3D geometric graph neural networks are not included in the comparison.

2. The models used in the study are essentially existing approaches, with no novel model proposed by the authors.

3. The analysis of the experimental results remains somewhat limited and could benefit from further depth.

Since this paper primarily focuses on datasets and benchmarks, the novelty of the methods is not as critical. The authors have also provided additional experiments with more recent models and detailed explanations of these results. Following the rebuttal, all reviewers reached a consensus and expressed positive feedback about the submission. Therefore, I recommend acceptance.

**Additional Comments On Reviewer Discussion:**

During the rebuttal period, the authors addressed the following points:

In response to concerns from Reviewer JSSw and HBUE, the authors provided additional experiments with recent models, which satisfied both reviewers.

Regarding the limited methodological innovation, as pointed out by Reviewer JSSw and k6pU, the authors clarified that the paper focuses on "datasets and benchmarks," which still offers valuable contributions to the field. Both reviewers expressed satisfaction with these explanations.

Reviewer HBUE raised additional concerns about including more baselines (high-degree models) and FastEGNN settings, which the authors have adequately addressed.

Overall, Reviewers JSSw and HBUE were actively engaged during the author-reviewer discussion period, while Reviewer k6pU did not participate but agreed to raise the score during the AC-reviewer discussion phase. In summary, all concerns were addressed, and all reviewers gave positive feedback on the revised submission.

---

### Decision · Program_Chairs · 2025-01-22

Accept (Poster)